# Proteomics of colorectal tumors identifies the role of CAVIN1 in tumor relapse

Ana Martinez-Val [ID][1,2,7✉], Leander Van der Hoeven [ID][1], Dorte B Bekker-Jensen[1,8], Margarita Melnikova Jørgensen[3,4,9], Jesper Nors [ID][5,6], Giulia Franciosa [ID][1✉], Claus L Andersen [ID][5,6✉], Jesper B Bramsen [ID][5,6✉] & Jesper V Olsen [ID][1✉]

## Abstract

Colorectal cancer molecular signatures derived from omics data can be employed to stratify CRC patients and aid decisions about therapies or evaluate prognostic outcome. However, molecular biomarkers for identification of patients at increased risk of disease relapse are currently lacking. Here, we present a comprehensive multi-omics analysis of a Danish colorectal cancer tumor cohort composed of 412 biopsies from tumors of 371 patients diagnosed at TNM stage II or III. From mass spectrometry-based patient proteome profiles, we classified the tumors into four molecular subtypes, including a mesenchymal-like subtype. As the mesenchymal-rich tumors are known to represent the most invasive and metastatic phenotype, we focused on the protein signature defining this subtype to evaluate their potential as relapse risk markers. Among signature-specific proteins, we followed-up Caveolae-Associated Protein-1 (CAVIN1) and demonstrated its role in tumor progression in a 3D in vitro model of colorectal cancer. Compared to previous omics analyses of CRC, our multi-omics classification provided deeper insights into EMT in cancer cells with stronger correlations with risk of relapse.

**Keywords** Colorectal Cancer; Proteomics; Tumor Relapse; CAVIN1
**Subject Categories** Cancer; Proteomics

## Introduction

Colorectal cancer (CRC) is the third most commonly diagnosed cancer worldwide and the second leading cause of cancer death globally. CRC is caused by abnormally and uncontrollably growing cells in the lining of the colon or rectum, where they can form tumors that can metastasize to other parts of the body. CRC is typically staged from I to IV following the tumor, node, and metastasis (TNM) system, which helps doctors diagnose the extent of the cancer and plan appropriate treatment. In stage II, the cancer has grown through the muscle layer of the colon or rectum but it has not spread to nearby lymph nodes, while in stage III is when the cancer has spread to nearby lymph nodes, but it has not spread to distant organs. Finally, stage IV is when the cancer has spread to distant organs, such as the liver or lungs. Once diagnosed, treatment for colon cancer in pre-metastatic stages is surgery, but it can be accompanied by adjuvant chemotherapy in certain cases. However, approximately one-third of patients which underwent surgical resection of the tumors eventually relapse (Schellenberg et al, 2022), and little is known about the molecular mechanisms behind the risk of relapse. Relapse could be due to remnant tumor cells not excised at the time of surgery or due to the presence of metastasis that has not been yet diagnosed. Importantly, the most relevant prognostic factor of risk of relapse is the stage of primary CRC (Zare-Bandamiri et al, 2017). However, while microsatellite instability (MSI) is a marker of lower risk of relapse, the number of lymph nodes, vascular and perineural invasion have been shown to be predictors of metastasis (Guraya, 2019). To minimize relapse cases, it is crucial to identify patients with aggressive disease and at high risk of relapse and provide them with additional treatment. Since CRC comprises a group of neoplastic diseases with diverse underlying molecular mechanisms, resulting in differential therapy response and prognosis, molecular subtyping offers a valuable approach for effective patient stratification. So far, this has primarily been achieved through transcriptomics-based classification, with the consensus molecular subtypes (CMS) being the most popular (Guinney et al, 2015). The CMS classification divides colon cancer into four major subtypes: CMS1 to CMS4. CMS1 (immune subtype) is characterized by being hypermutated, with high microsatellite instable (MSI-h) status and high immune cell infiltration. CMS2 (canonical subtype) is characterized as being epithelial-rich with Wnt and MYC pathway activation and high proliferation. CMS3 (metabolic subtype) is characterized by metabolic dysregulation and upregulation of genes related to lipid metabolism, and CMS4 (mesenchymal subtype) is characterized by stromal invasion, extracellular matrix remodeling and angiogenesis.

[1]Novo Nordisk Foundation Center for Protein Research, Department of Cellular and Molecular Medicine, Faculty of Health and Medical Sciences, University of Copenhagen, Copenhagen, Denmark. [2]CIBER de Enfermedades Cardiovasculares (CIBERCV), Madrid, Spain. [3]Institute of Pathology, Randers Regional Hospital, Randers, Denmark. [4]Department of Pathology, Aalborg University Hospital, Aalborg, Denmark. [5]Department of Clinical Medicine, Aarhus University Hospital, Aarhus, Denmark. [6]Department of Molecular Medicine, Aarhus University Hospital, Aarhus, Denmark. [7]Present address: Cardiovascular Proteomics Laboratory, Centro Nacional de Investigaciones Cardiovasculares Carlos III (CNIC), Madrid, Spain. [8]Present address: Evosep Biosystems, Odense, Denmark. [9]Present address: Department of Clinical Medicine, Aalborg University, Aalborg, Denmark. ✉E-mail: ana.martinezdelval@cnic.es; giulia.franciosa@cpr.ku.dk; cla@clin.au.dk; bramsen@clin.au.dk; jesper.olsen@cpr.ku.dk

Molecular subtyping has enabled more targeted therapeutic approaches. For instance, immunotherapy is emerging as a promising treatment option for CMS1 patients with MSI-h tumors. These MSI-h tumors have been found to respond poorly to adjuvant therapy with 5-fluorouracil (Sargent et al, 2010) but are associated with an immunologically active tumor microenvironment (Wu et al, 2023), making them more sensitive to immunotherapy, such as anti-PD-1 therapy (Le et al, 2015). Conversely, tumors with high stromal content, characteristic of the CMS4 subtype, are associated with poor prognosis and resistance to treatment (Isella et al, 2015; Calon et al, 2015), highlighting the need for new targeted therapies for this subtype.

Another emerging promising approach for molecular subtyping is the use of mass spectrometry (MS)-based proteomics, or the so-called multi-omics approach that combines several omics technologies (Zhang et al, 2014a; Li et al, 2020; Joanito et al, 2022; Kong et al, 2023; Vasaikar et al, 2019).

In this study, we present a large-scale multi-omics analysis of a Danish colorectal cancer tumor cohort composed of 412 biopsies from tumors of patients diagnosed at pre-metastatic stages (II and III). Based solely on proteomics data, we could recapitulate the known CMS transcriptomics-based subtypes, identifying protein-centric signatures for each one of them: epithelial-to-mesenchymal transition (EMT)-like and stroma-rich subtype, metabolic subtype, canonical subtype and immune subtype. We focused on the proteins that were enriched in the EMT-like subtype and evaluated their link to relapse. State-of-the-art mass spectrometry-based proteomics of the CRC tumors allowed us to complement the information derived previously by transcriptomics, especially for the CMS4/EMT-like subtype.

We focused on the proteins that were enriched in this subtype and evaluated their link to relapse. In this work, relapse is defined as local relapse when it occurs at the same place as the original cancer (at the anastomosis, or in the tissue and lymph nodes in the region around the anastomosis) or as distant relapse, if it occurs in organs or tissues far from the original cancer.

Among these proteins, we focused on Cavin1, which was not only over-expressed in EMT-like tumors, but displayed higher abundance in the entire population of stage III tumor patients that suffered relapse. To investigate the molecular bases of Cavin1 role in CRC recurrence, we made use of a 3D in vitro model, and showed that Cavin1 is expressed by epithelial cancer cells and involved in both tumor cells' proliferation and invasion. Importantly, CAVIN1 has previously been studied in the context of tumor progression, and cancer cells expressing higher levels of this protein, as well as other caveolin proteins, tend to be more aggressive and metastatic. This higher metastatic potential has been associated with increased capacity for anchorage-independent growth (Gupta et al, 2014). In this regard, CAVIN1 and other caveolae-associated proteins have been described as markers of poor prognosis in pancreatic cancer (where they modulated caveolin-1 function (Liu et al, 2014)), glioblastoma (Pu et al, 2019), breast (Mercier and Lisanti, 2012), or rhabdomyopsarcoae (Faggi et al, 2015).

Finally, using a second cohort of higher relapse-risk patients, we performed hybrid-DIA phosphoproteomics analysis for high-sensitivity detection of aberrant phosphorylation-dependent signaling pathways (Bhullar et al, 2018). Interestingly, our analysis identified down-regulation of the mTOR signaling pathway (Francipane and Lagasse,

2014; Brandt et al, 2018) in patients in risk of relapse, which anti-correlated with CAVIN1 levels in tumors of the EMT-like phenotype.

# Results

## Study design and pre-metastatic colorectal cancer patient cohort

This study comprises two patient cohorts: one large discovery cohort and a second smaller cohort for validation and phospho-proteomics analysis. The first cohort comprised 361 tumor biopsies from 321 patients treated with curative intended surgery for stage II or III colorectal cancer (CRC). The analyzed biopsies included 197 samples from stage II tumors and 164 samples from stage III (Fig. 1A). Fifty additional patients were selected for the second cohort based on their higher frequency of relapse (distant and/or local) after surgery or the probability of relapse predicted by the LASH algorithm (Lash et al, 2015). Basic demographic and tumor characteristics are shown in Fig. 1B, including microsatellite status. At surgery, all patients were non-metastatic, however, despite surgery eliminating all visible disease (R0 resections), 24% of patients were later diagnosed with metachronous metastatic relapse (Fig. 1B and Dataset EV1). Importantly, since the primary treatment for all included patients were curative intent surgery, all patients were treatment-naïve prior to surgery. Accordingly, any relapse event must mean that residual disease persisted in the patient after surgery. In the case of local relapse then it can be argued that surgical removal was incomplete. In case of distant relapse, it can be argued that surgical removal of the primary tumor was complete, but that occult metastatic spreading (undetectable by diagnostic CT imaging) had already occurred at time of surgery, and that this spread later manifested as detectable lesions.

Moreover, CRC patients in the cohorts were assessed for microsatellite stability after diagnosis and separated into microsatellite instable (MSI) or microsatellite stable (MSS) (Gupta et al, 2018). MSS/MSI status as well as tumor stage were the two clinical traits linked to risk of relapse in the discovery cohort, while other clinical characteristics, such as tumor location or demographic variables such as age or sex did not show statistically significant connections with higher risk of relapse in these patients (Fig. 1C).

## Rapid proteomics profiling identifies four distinct molecular subtypes and reveals differences with transcriptomics regulation

We performed rapid proteomic profiling of tumor tissue resected at the time of surgery by using high-performance liquid chromatography tandem mass spectrometry (LC-MS/MS). Our aim was to explore the molecular characteristics of colorectal cancer and identify tumor proteome profiles associated with increased risk of metachronous metastasis.

To facilitate fast and global proteome profiling of the clinical samples, we designed a data-independent acquisition (DIA) strategy by which each tumor sample digest was analyzed in duplicates using short 21-min online LC gradients in Evosep One in combination with High Field Asymmetric Waveform Ion Mobility Spectrometry (FAIMS) in an Orbitrap Exploris 480 mass spectrometer (Bekker-Jensen et al, 2020b). The resulting

## A

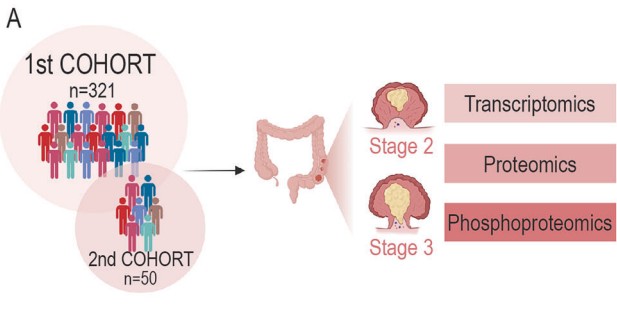

## B

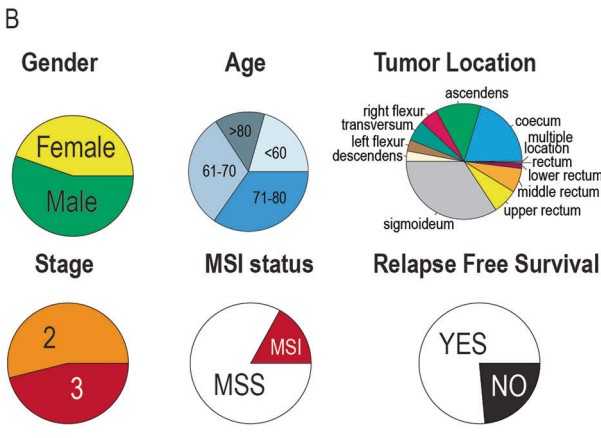

## C

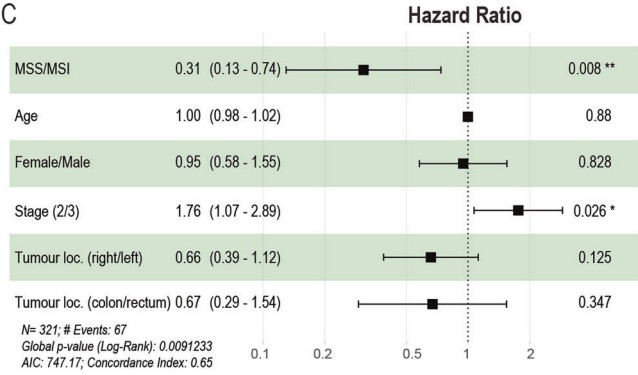

**Figure 1. Colorectal tumor cohort description.**

(A) Two cohorts were employed in this project, a discovery cohort of 361 patients, and a second validation cohort of 51 patients. Colorectal tumors from all patients were removed during surgery, and classified as stage II or III. Quantitative proteomics profiling was performed for all samples, while transcriptomics was performed only in 1st cohort and phosphoproteomics of the 2nd cohort. (B) Pie chart showing the distribution of relevant traits (gender, age, tumor location, TNM stage, MSI status and Relapse Free Survival) for the full cohort. (C) Forest plots of Cox proportional-hazards regression analysis of relapse risk for clinical and demographic characteristics in the first cohort ($n = 321$ patients). For each trait, it is indicated the hazard ratio, the confidence interval (95% of it is represent as the error bar) and the exact p-value. The symbol '*' indicates p-value < 0.05 and '**' indicates p-value < 0.01.

raw LC-MS/MS files were analyzed by using library-free directDIA searches in Spectronaut (v16) covering almost 8000 protein groups in total. After quality control filtering involving removal of LC-MS/MS runs of poor quality and requiring identified protein-coding-genes to be quantified in at least 75% samples (see Methods), the proteome profiles of colorectal samples consisted of 4964 quantified

protein-coding-genes (Dataset EV2). Importantly, the cohort was comprised of 9 batches, collected and processed for mass spectrometric analysis at different dates, reflecting one of the main challenges of a continuous sample collection in proteomics. The difference between collection, reception and processing of the samples implied that two different sample preparation workflows were employed: conventional in-solution trypsin digestion (batches 1 to 3) and Protein Aggregation Capture (PAC) on-bead digestion (Batth et al, 2019) (batches 4 to 9) (Fig. EV1A, Dataset EV2). Quality control samples, comprised of 100 ng of HeLa peptide tryptic digest, were analyzed together with the colorectal cancer samples to ensure MS-reproducibility across batches (Fig. EV1B). However, batch-effect correction and normalization were required to unify between to the two sample processing protocols and different acquisition time points. Principal Component Analysis (PCA) after batch-correction showed no separation based on clinical traits (MSI status, relapse-free survival and tumor location), demographical characteristics (age, sex) or experimental batches (Fig. EV1C,D). Finally, to account for the heterogeneous nature of tumors, multiple biopsies were collected for some of the patients, which allowed us to evaluate the intra- and inter-patient variability of tumors. Here, we observed that protein correlation was higher in intra-patient biopsies than among inter-patient biopsies, reflecting a patient-specific primary tumor proteomics signature (Appendix Fig. S1).

Consensus clustering analysis of the tumor proteome profiles was applied to stratify samples into four proteomics (PROT) subtypes (Dataset EV1 and Fig. 2A). PROT subtype 4 showed strong enrichment of patients with MSI-high status (Fig. 2A). In contrast, TNM stage was evenly distributed across PROT subtypes 1, 2 and 4 ranging between 44.7 to 55.4%. However, in PROT subtype 3, the proportion of TNM Stage 3 patients was only 29.5%. Contrariwise, the proportion of patients that suffered relapsed only varied slightly between subtypes: 26.2% in PROT 3, 25.7% in PROT 1, 18.4% in PROT 2, and 17.6% in PROT 4. In PROT Subtype 1, 84.2% of the patients in TNM Stage 3 patients suffered relapse, while, in contrast, that proportion was significantly lower in the other PROT subtypes (Fig. 2B).

ESTIMATE score analysis (Yoshihara et al, 2013) revealed a significant stromal enrichment in PROT subtype 1, and higher immune infiltration in PROT subtype 4 (Fig. EV2A). In agreement with higher stromal and immune content, PROT 1 and 4 showed lower tumor purity than PROT 2 and 3 (Fig. 2A). The proteomics-derived subtypes generally correlated well with some of the CMS-based subtypes obtained from the RNASeq analysis of the same samples (Figs. 2A and EV2B). For example, the PROT subtype 1 showing higher stromal enrichment, matched the CMS4 subtype in the transcriptomic data. Likewise, the PROT subtype 4, which included most MSI-high patients and showed higher immune scores, correlated mainly with the CMS1 subtype. In contrast, PROT subtype 2 is grouped mainly with the CMS3 samples, but also a significant proportion of the CMS2 samples. Finally, PROT subtype 3 included mainly CMS2 samples. The distribution of CMS2 samples within two PROT subtypes might indicate a divergence between the transcriptomic-based CMS classification and the protein-based signatures. Tissue cellular composition must be taken into account when analyzing bulk omics data. The derived signals reflect the mixture of malignant proliferating cells and numerous distinct non-cancerous cell types that comprise the

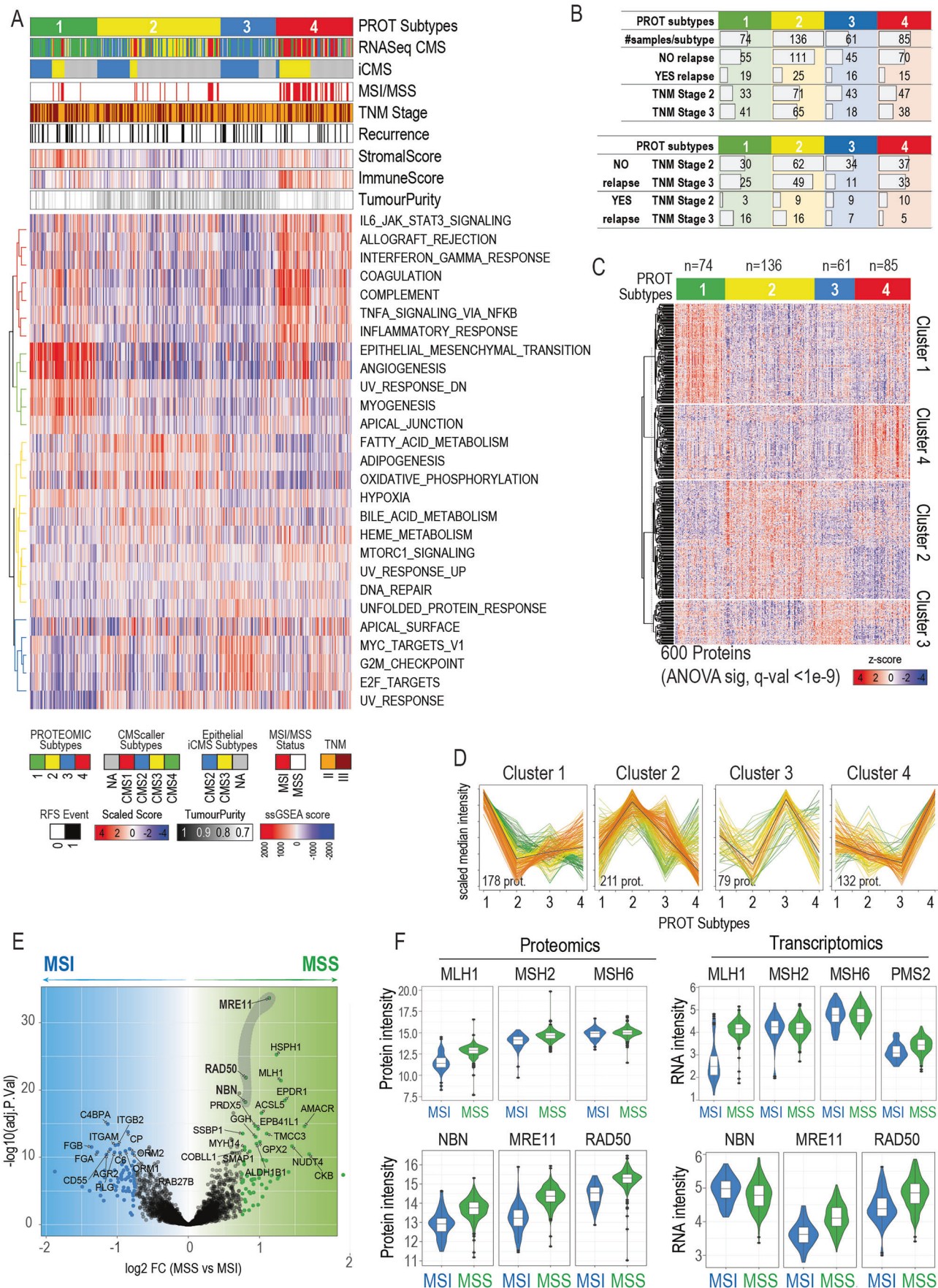

**Figure 2. Characterization of proteomics-based molecular subtypes.**

(A) (Top) Patient sample classification based on consensus clustering using their proteomics signatures, and CMS subtypes assigned to the same patients using RNASeq signatures, as well as iCMS epithelial subtype assignment using Nearest Template Prediction with proteomics data. Below, clinical traits for MSI status, TNM stage and relapse. Lastly, scaled values of Stromal and Immune Score, as well as Tumor purity. (Bottom) Heatmap of ssGSEA enrichment scores of samples in upper panel. (B) Table with number of samples in each proteomics subtype, as well as the number of samples depending of the relapse status and TNM stage. (C) Heatmap of significant proteins differential between proteomics subtypes (ANOVA, FDR Benjamini-Hochberg). Proteins were grouped into 4 clusters (Euclidean distance). Protein intensity was scaled across patients prior to performing the clustering. (D) Profile plots of the proteins in each cluster from (A). Protein intensities were collapsed to their median value across patients in each proteomics subtype, then scaled across subtypes. Black line indicates the median centroid of the distribution. Color gradients indicate the spearman correlation to that centroid (green = low correlation, red = high correlation). (E) Volcano plot of the log2 fold-change of proteins measure in MSS ($n = 291$) compared to MSI ($n = 65$) patients, plotted versus the $-\log10$ adjusted $p$-value. Results were obtained using limma two-sided t-test with FDR corrected by Benjamini-Hochberg. In green and blue, proteins with fold-change >0.75 (in log2) and FDR adjusted $p$-value < 0.01. Dots labeled are proteins with fold-change >0.75 (log2) and adjusted $p$-value < 1e−5. (F) Boxplot of protein (left) and mRNA (right) levels of proteins from the MMR complex (top) and NBN complex (bottom). For proteomics, MSI $n = 65$, MSS = 291. Boxplot limits indicate the 25th and 75th percentiles as determined by R software; whiskers extend 1.5 times the interquartile range from the 25th and 75th percentiles, outliers are represented by dots.

tumor microenvironment and that have influence in the drug response and the development of the disease (Li et al, 2022). Even though only single-cell strategies can directly account for the tissue heterogeneity, in silico annotation strategies such as xCell (Aran et al, 2017) can digitally dissect the tissue diversity from bulk omics data. Consequently, we employed xCell to annotate our RNASeq dataset and, in agreement with our previous results, we identified distinct cell types in each subtype, highlighting an overrepresentation of immune cell types in subtype 4 and of mesenchymal cell types in subtype 1 (Fig. EV2C).

Given the similarities between patient stratification based on transcriptomic and proteomic data, we decided to evaluate whether both classifications had the same prognostic potential. When analyzing all samples together (including both stage II and III patients) (Appendix Fig. S2A,B), neither of the proteomic clusters were associated with recurrence. In contrast, the RNA-Seq-derived groups showed that patients classified in the CMS1 group had a lower recurrence rate compared to the others. However, when we focused on stage III patients, we observed that the proteomic classification, specifically the PROT subtype 1, was associated with a higher probability of recurrence, while this was not the case for patients classified as CMS4, the molecular equivalent based on RNA-Seq data (Appendix Fig. S2C,D).

To complement the CMS classification system and include the information derived from epithelial cell diversity, Joanito et al has defined two epithelial-based subtypes (iCMS2 and iCMS3) based on single-cell transcriptomic analysis (Joanito et al, 2022). We employed nearest template prediction (Hoshida, 2010) and the iCMS2 and iCMS3 signatures provided to stratify the proteomics data into these two novel subtypes (Fig. 2A). Although many samples could not be classified into either subtype, we found that the PROT subtype 4 grouped mainly iCMS3 samples, which is in line with the findings of Joanito and colleagues that indicate that iCMS3 mostly group MSI-high patients. In contrast, PROT subtype 3 included only iCMS2 samples, while both subtypes 1 and 2 showed a mixed distribution of iCMS2 and iCMS3 subtypes, respectively. Moreover, iCMS2 samples from PROT subtype 2 matched the samples classified as CMS2 from RNASeq data (Fig. 2A).

Next, to annotate the molecular functions driving the proteomics-based tumor stratification, we performed single-sample Gene Set Enrichment Analysis (ssGSEA) with cancer Hallmark gene signatures. Differentially regulated gene sets

between the four PROT subtypes were visualized in a heatmap (Fig. 2A). PROT subtype 1 was overrepresented in terms such as "Epithelial to mesenchymal transition" and "Angiogenesis". Moreover, confirming the immune-activated nature of PROT subtype 4, this group of samples was enriched in terms related to immune and inflammatory responses. Conversely, PROT subtype 2 enrichment reflected a more metabolically active type of tumors, whereas PROT subtype 3 showed enrichment in terms connected to the canonical signature of colorectal cancer (Fig. 2A). Moreover, ANOVA analysis identified 600 proteins differentially regulated among the subtypes (BH FDR-adjusted $p$-value < 1e−9) (Dataset EV2 and Fig. 2C). Clustering analysis of the protein profiles resulted in four protein clusters, each one with a characteristic proteomic profile that can be linked to the proteomic subtypes defined before (Dataset EV2 and Fig. 2D).

In line with the stromal-rich nature of PROT subtype 1 samples, we found extracellular matrix proteins such as collagens (Vellinga et al, 2016) and lumican (LUM) (Zang et al, 2021) to be specifically upregulated in this group. Moreover, we found novel protein markers for EMT such as LAMB2, TNXB, DCN and CAVIN1 (Huang et al, 2022). PROT subtype 2 is mostly enriched in metabolic enzymes. EPDR1, a known marker of relapse risk, was included in the PROT subtype 3 (Gimeno-Valiente et al, 2020). Interestingly, the MRN complex (MRE11, RAD50 and NBN) was also part of our PROT subtype 3. The MRN complex plays a critical role in cellular response to DNA damage and maintenance of chromosome integrity, and it is known to be differentially regulated between MSS and MSI patients. Finally, also in line with the immune-activated and MSI-high profile of subtype 4, we encountered immune-related proteins upregulated in this subtype, such as fibrinogens (FGG, FGB, FGA) (Parisi et al, 2022) and members of the complement system (C3, C4A, C5, C9) (Dataset EV2). Gene ontology (GO) enrichment analysis of the proteins in each ANOVA-defined clusters (Fig. EV2D) confirmed the molecular characteristics observed by Hallmark signatures in the proteomics subtypes (Fig. 2A).

Moreover, we compared the proteomic subtypes defined in this publication with similar classifications previously reported by Zhang and colleagues (Zhang et al, 2014b), where they identified five molecular subtypes. Zhang's subtype C, classified as the EMT-type, showed the greatest overlap with our PROT 1 subtype, although it also shared some overlap with proteins from the PROT 4 subtype. Interestingly, although Zhang's subtype B did not

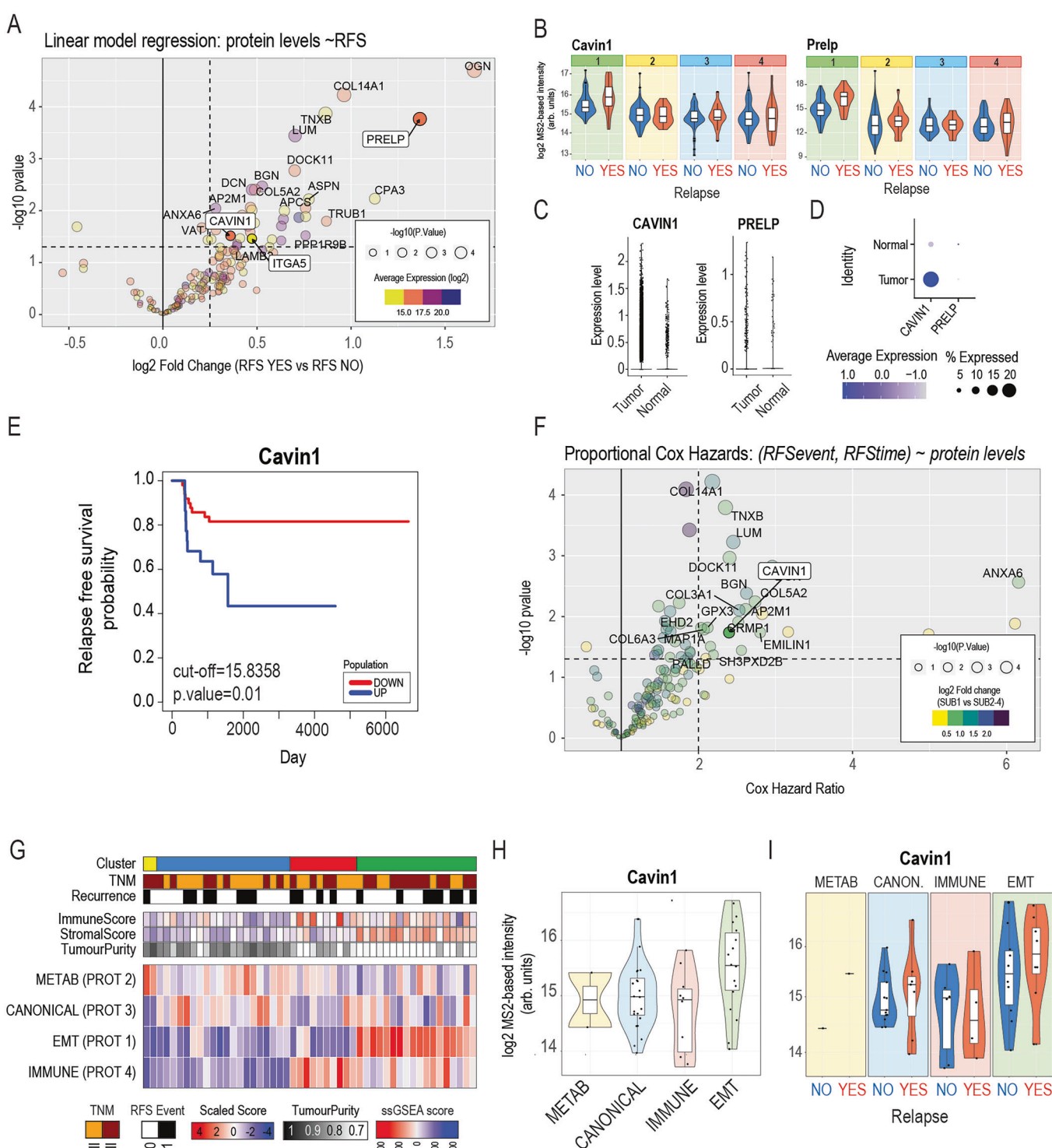

overlap with PROT subtype 4 (Appendix Fig. S3A), performing ssGSEA on our proteomic data using Zhang et al's signatures revealed a strong correlation with PROT 4 patients (Appendix Fig. S3B). Conversely, only Zhang's subtype E overlapped with the other two PROT subtypes, leaving subtypes A and D unrepresented in the proteomic signatures defined here, which is consistent with the Unified Molecular Subtypes more recently described by Vasaikar

et al (Vasaikar et al, 2019). To extend the validity of our proteomic subtypes, we applied them to a published cohort by Li et al (Li et al, 2023), which evaluated the effects of chemotherapy on relapse in colorectal cancer patients. When applying our proteomic subtype signatures to the Li et al dataset, we observed that the PROT subtype 1 signature could significantly distinguish patients who later developed metastasis (Appendix Fig. S3C). However, we were

**Figure 3.  Prognostic markers assessment in proteomics subtype 1.**

(A) Volcano plot of limma results (moderated t-test) comparing protein levels of PROT 1 subtype signature ($n = 144$ proteins) between positive relapse and negative relapse events in patients classified in this subtype ($n = 71$ patients, 74 samples). Color encodes the average protein level. Size encodes the statistical significance. Highlighted in white, the proteins with prognostic potential in CRC based on TCGA data from HPA. (B) Boxplot of CAVIN1 and PRELP protein levels (log2, arbitrary units) in the four PROT subtypes, grouped by the relapse status. Subtype 1, relapse YES $n = 19$; Subtype 1, relapse NO $n = 55$; Subtype 2, relapse YES $n = 25$; Subtype 2, relapse NO $n = 111$; Subtype 3, relapse YES $n = 16$; Subtype 3, relapse NO $n = 45$; Subtype 4, relapse YES $n = 15$, Subtype 4, relapse NO $n = 70$. (C, D) Expression level of CAVIN1 and PRELP in tumor cells or non-tumor cells retrieved from Joanito et al single cell dataset. (E) Kaplan–Meier curves for relapse-free survival of patients based on CAVIN1 levels. CAVIN UP $n = 22$, CAVIN DOWN $n = 49$. Statistical significance was calculated using a log-rank test. (F) Results form a Proportional Cox Hazards test results (Relapse yes $n = 19$, Relapse no $n = 55$) log2 hazard ratios (HR) against the $-$log10 p-values obtained for each protein marker of subtype 1 ($n = 144$ proteins). Color indicates the abundance ratio (in log2) between the levels of the protein in subtype 1 and the other subtypes. Labeled dots represent proteins more abundant in subtype 1 (log2 ratio >0.5) with a HR > 2. Squared labels highlight those proteins more abundant in subtype 1 (log2 ratio >0.5), with HR > 2 that are classified as poor prognosis markers in colorectal cancer in the Human Protein Atlas. (G) Heatmap of ssGSEA enrichment scores from PROT subtype signatures of samples in the second cohort. PROT 1 = green, PROT 2 = yellow, PROT 3 = blue, PROT 4 = red. (H) Boxplot of CAVIN1 protein levels (log2, arbitrary units) in the four patient subtypes from the second cohort. (I) Boxplot of CAVIN1 protein levels (log2, arbitrary units) in the four patient subtypes from the second cohort grouped by relapse event. METAB. subtype $n = 2$ (RFS Yes $n = 1$, RFS No $n = 1$), CANON. subtype $n = 20$ (RFS Yes $n = 7$, RFS No $n = 13$), IMMUNE subtype $n = 10$ (RFS Yes $n = 8$, RFS No $n = 10$). For (B), (H) and (I), boxplot limits indicate the 25th and 75th percentiles as determined by R software; whiskers extend 1.5 times the interquartile range from the 25th and 75th percentiles, outliers are represented by dots. Source data are available online for this figure.

unable to attribute the higher relapse risk to any specific therapy evaluated in that study (Appendix Fig. S3D).

Importantly, the multivariable Cox proportional-hazards regression analysis performed on the clinical and demographical traits of this cohort pointed to MSS and MSI status as linked to relapse risk (Fig. 1C). In line with this, previous studies have shown that MSI status is linked to better prognosis (Popat et al, 2005). Our proteomics profiling reflected that microsatellite stability status is a highly relevant characteristic for patient stratification, as shown by 66% of the MSI patients are specific to PROT Subtype 4 (hypergeometric test: p-value < 2.7e−16).

From a molecular perspective, MSI-high status is characterized by defects in the DNA mismatch repair machinery (PSM2, MSH6, MLH1 and MSH2 proteins) leading to higher mutation rate in these tumors. Differential expression analysis between MSS and MSI patients reflected a higher expression of immune-related proteins such as FGB, FGG, and FGA and of immune activation and inflammatory pathways (Fig. 2E and Dataset EV3). Also, we could identify established protein markers for MSS/MSI status, such as lower expression at protein and transcript level of the MMR component MLH1 in MSI patients (Fig. 2F). This supports previous research that points DNA methylation associated with transcriptional silencing of MLH1 as the underlying cause of MMR defects in most sporadic colorectal cancers with MSI (Herman et al, 1998). On the other hand, other complexes involved in DNA repair, such as the MRN complex that are down-regulated in MSI-high tumors (Miquel et al, 2007; Gao et al, 2008). Interestingly, all three components of this complex (MRE11, RAD50, and NBN) showed statistically significant higher protein abundance in MSS than in MSI patients (Fig. 2E,F). However, only RAD50 and MRE11 transcripts were down-regulated in MSI samples, while NBN did not follow the same trend (Fig. 2F). This demonstrates that although our data showed a good correlation between proteomics and transcriptomics when grouping MSI-high patients, proteomics revealed novel information about potential post-translational regulatory mechanisms.

Finally, to enrich the molecular annotation of the proteomics-derived data, we employed Weighted Gene Coexpression Network Analysis (WGCNA) (Langfelder and Horvath, 2008) to explore potential relationships between protein modules and clinical traits. WGCNA revealed two separate protein modules (black and purple)

that correlated with MSI status (Appendix Fig. S4A). One protein module ("black module") correlated with both MSI status and immune score. In contrast, the other module ("purple module") only show positive association with MSI status but not with immune infiltration. Gene Ontology (GO) annotation of the proteins from those modules confirmed that the black module was enriched in proteins involved with immune response, whereas the purple module enriched terms related to DNA processing (Appendix Fig. S4B). These results indicate that the proteomics signature can be used to stratify MSI-high patients into those with high and low immune-score as an indicator of conceivable responsiveness to immunotherapies (Becht et al, 2016; Wang et al, 2023b; Borelli et al, 2022).

## Identification of Cavin1 as an epithelial to mesenchymal transition marker with prognostic value in PROT subtype 1 CRC patients

Relapse-free survival (Appendix Fig. S5A) as well as multivariable Cox regression analysis of the association between clinical traits and the risk of relapse (Fig. 1C) indicated that TNM tumor stage is associated to relapse risk in colorectal cancer patients. Comparison of TNM stage II versus TNM stage III colorectal cancer proteomics profiles revealed that stage III tumors are enriched in proteins related to "Epithelial to mesenchymal transition" (Appendix Fig. S5B–D). This finding agrees with previous reports that link EMT-rich cancer subtypes to poor prognosis (Calon et al, 2015; Isella et al, 2015). The proteomics-based subtype 1 defined in here showed significant overrepresentation of proteins from the EMT pathway (Fig. 2A,B), as well as higher stromal score (Fig. 2A).

Using Kaplan–Meier analysis to evaluate relapse-free survival, neither of the proteomics subtypes showed higher risk of relapse (Appendix Fig. S5B). However, PROT subtype 1 had a higher proportion of relapse cases among the TNM Stage III tumors (Fig. 2B). Therefore, we decided to explore further the proteins within this subtype to identify markers that could indicate the link between EMT and colorectal cancer relapse. In this regard, we focused on PROT subtype 1 patients and performed differential expression analysis based on their relapse status (Fig. 3A) on the PROT subtype 1 signature proteins (Fig. 2C,D). Overall, there was a trend on this subgroup of proteins towards higher values in relapse

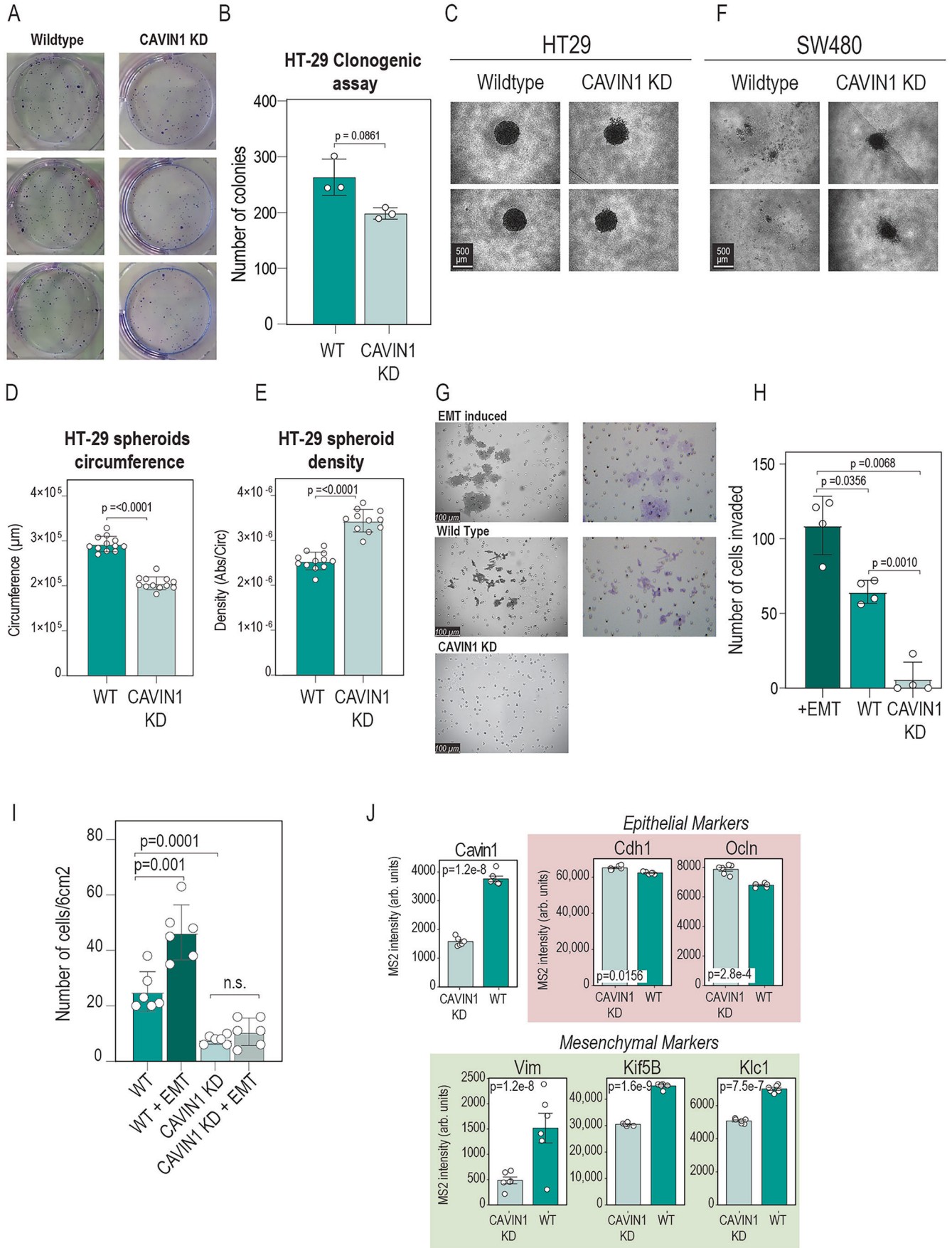

◀ **Figure 4. In vitro validation of EMT-marker CAVIN1 relation with tumor invasiveness.**

(A) Clonogenic assay images from HT29 WT and CAVIN1-KD ($n = 3$ biological replicates). (B) Number of colonies obtained from clonogenic assay in (A). (C) Representative image of spheroids derived from 7000 cells of HT29 cell line and grown for 96 h, either for Wild Type (left) or CAVIN1 knockdown (right) ($n = 4$ biological replicates). (D) Quantification of total circumference length of wild type or CAVIN1 knock-down HT-29 derived spheroids grown for 96 h ($n = 12$). P-value $= 2.384e−12$. (E) Quantification of cell density (measured as the ratio of the absorbance at 450 nm from CCK-8 test vs the circumference of the spheroid) of wild type or CAVIN1 knock-down HT-29 derived spheroids grown for 96 h ($n = 12$ biological replicates). P-value $= 5.895e−09$. (F) Representative image of spheroids derived from 7,000 cells of SW480 cell line and grown for 96 h, either for Wild Type (left) or CAVIN1 knockdown (right) ($n = 4$ biological replicates). (G) Representative images of cell invasion assay in EMT-induced conditions, wild type and CAVIN1 knockdown. (H) Quantification results for the invasion assay of HT-29 derived spheroids in EMT-induced condition, wild type and CAVIN1 knockdown. Invasion potential is measured as the number of cells that migrated through the matrigel membrane after 24 h ($n = 4$ biological replicates). (I) Quantification results for the invasion assay of HT-29 adherent cells, either wild type (control) or CAVIN1 KD in standard or EMT-induced conditions. Invasion potential is measured as the number of cells that migrated through the matrigel membrane after 24 h ($n = 6$ biological replicates). For (A–I), the statistical test used was a Welch two-samples t-test. (J) Mass-spectrometry-based proteomics abundance of CAVIN1 in HT-29 wild type cells or CAVIN1 KD cells, as well as relevant mesenchymal and epithelial markers ($n = 6$ biological replicates). Statistical significance of the difference in abundance of each protein was calculated using limma two-sample test, with FDR correction by Benjamini-Hochberg. In all panels, the height of the bar represents the mean of the individual measurements, and the error bars the standard error of the mean. Source data are available online for this figure.

cases, but specifically 38 of the proteins were significant upregulated (log2 fold change $>0.25$ and p-value $< 0.05$) in PROT subtype 1 patients with relapse (Fig. 3A). Among those 38 proteins, three of them were annotated as prognostic in the TCGA CRC dataset (Fig. EV3): CAVIN1, PRELP, and ITGA5. CAVIN1 is a protein involved in caveolae formation and organization, whereas PRELP and ITGA5 are structural proteins involved in cell adhesion. We focused on PRELP and CAVIN1 as potential prognostic markers, and discarded ITGA5 due to its lower protein abundance (Fig. 3A). Both, PRELP and CAVIN1 prognostic potential is limited to patients from PROT 1 for which they showed higher levels in relapse cases (Fig. 3B).

The predictive value of CAVIN1 and PRELP for assessing relapse risk could simply be due to a higher stromal content in patients with poorer prognosis. In fact, PRELP and CAVIN1 are proteins mainly expressed in cells of mesenchymal origin, as shown in The Human Protein Atlas data (Uhlen et al, 2017) (Appendix Fig. S6A,B). This is in line with previous literature showing elevated expression of a mesenchymal signature and poor prognosis in CRC samples is mainly due to tumor-associated stromal cells rather than by epithelial tumor cells (Calon et al, 2015; Isella et al, 2015). However, there is also evidence that CRC tumor cells can undergo EMT leading to higher invasiveness potential (Ieda et al, 2019). Therefore, we decided to explore whether CAVIN1 had a role in epithelial tumors cells, in terms of tumor formation and invasion, or if it was just restricted to the tumor microenvironment. Single-cell transcriptomics data of colorectal tumors from Joanito et al (Joanito et al, 2022) showed evidence that CAVIN1, but not PRELP, is expressed in epithelial tumor cells (Fig. 3C,D), although expressed at lower levels than in mesenchymal cells (Appendix Fig. S7A–D).

Using the relapse predictive value from CAVIN1 derived from the previous analysis, we observed that relapse cases were also significantly differentiated in a Kaplan–Meier analysis (Fig. 3E). Next, we also confirmed the role of CAVIN1 in CRC relapse using a Cox proportional Hazard model to estimate the relapse hazard ratio as function of PROT subtype 1 marker levels in patients from that subtype (Fig. 3F). From our three initial markers (PRELP, ITGA5, CAVIN1), it was only CAVIN1 which protein abundance correlated with the onset of relapse.

To validate the usefulness of our proteomics signatures to stratify CRC patients based on their molecular characteristics, we next wanted to validate them in a second CRC patient cohort. This cohort was comprised of fifty additional patients that were selected based on their higher frequency of relapse (distal and local) after surgery or the probability of relapse predicted by the LASH algorithm (Lash et al, 2015). The proportion of patients that suffered relapse in this cohort was higher than in the first cohort: 40% of the patients in this cohort while only 21% of the patients in the initial cohort. Proteomics profiles from the samples in this cohort were collapsed using ssGSEA analysis (Fig. 3G) with PROT subtype signatures defined in this work (Fig. 2C,D). The PROT subtype 1 or EMT and the PROT subtype 4 or Immune signatures were the ones to group better patients from this cohort. In agreement with this, we found that the Stromal Score correlated with the PROT subtype 1 or "EMT signature". In contrast, the Immune Score distribution was not so clear, although it was higher in patients with higher PROT subtype 4 scores (Fig. 3G). Using this classification in the second cohort, we again observed that CAVIN1 was higher in patients with higher PROT subtype 1 (Fig. 3H), which also was higher in relapse in this group of patients (Fig. 3I). Altogether, our data support the hypothesis that CAVIN1 levels correlate with higher risk of relapse in CRC patients with an EMT classification, in two independent cohorts.

Corroborating with our previous results, there is growing evidence that modulation of CAVIN1 pathways in tumor cells can alleviate the risk of metastasis (Wang et al, 2023a; Huang et al, 2022; Liu et al, 2014). Therefore, we decided to explore further the role of intra-tumor CAVIN1 levels in promoting invasiveness in colorectal cancer cell lines.

We employed the HT29 colorectal cancer cell line, which is an epithelial CRC cell line and therefore has not gone through EMT like other mesenchymal-like CRC cell lines such as SW480 (Berg et al, 2017). Also, HT29 cell line was derived from an MSS tumor, which is a better model to test CAVIN1 levels and its link to cancer progression, since our PROT Subtype 1 was mainly comprised of MSS patients (Fig. 2A). Metastasis arises following evolutionary events during which cancer cells acquire the ability to escape from the tumor, disseminate, and grow in distant organs. Therefore, we evaluated the colony forming capacity of HT29 wild type (WT) cell lines and how it is affected by the knock-down of CAVIN1 (CAVIN1-KD) and observed that CAVIN1-KD slightly decrease the capacity of HT29 cells to form colonies (Fig. 4A,B).

Following this, we grew multicellular spheroids from WT and CAVIN1-KD HT29 cells. CAVIN1-KD did not impair the formation of spheroids but, remarkably, we observed that

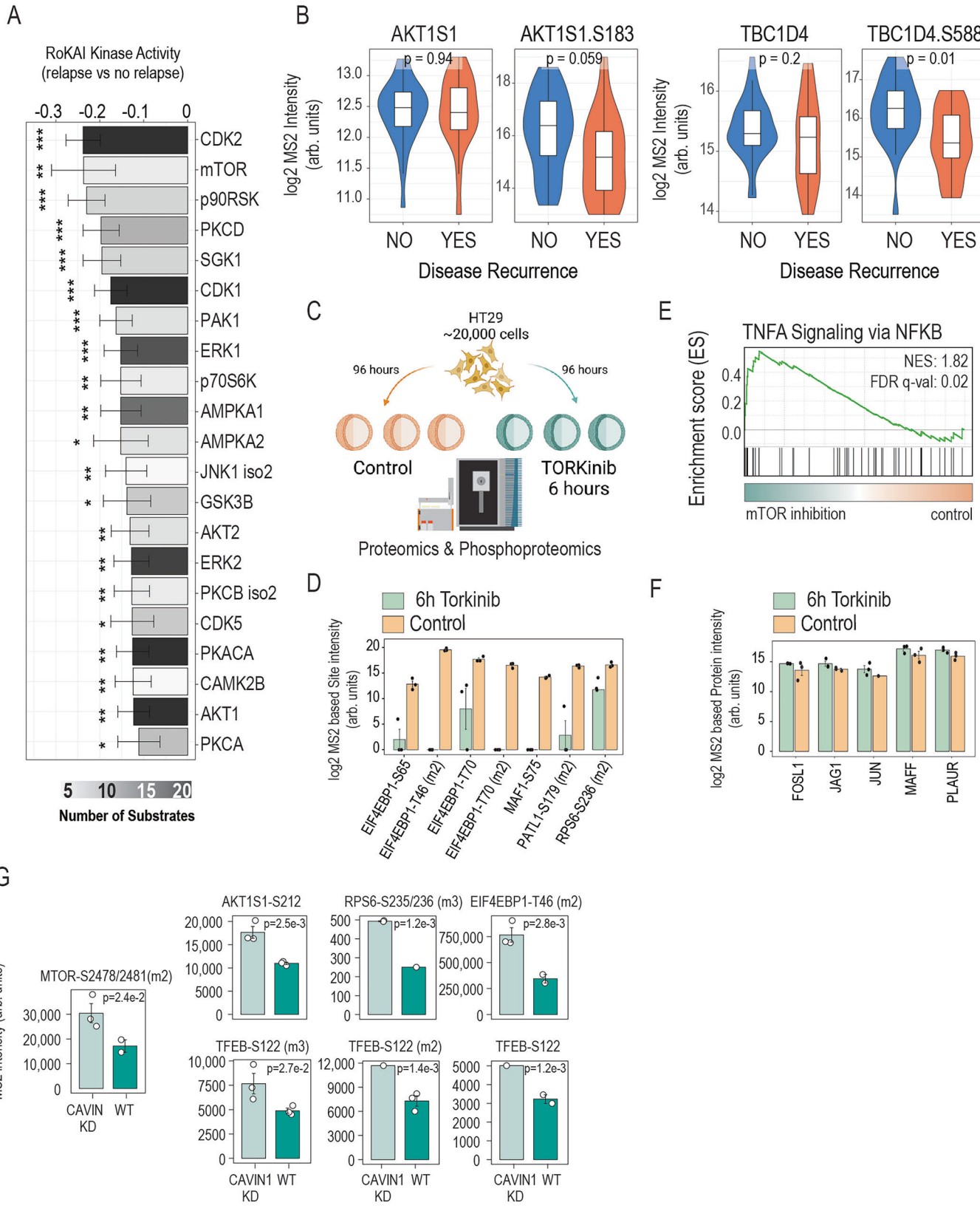

**Figure 5.  Phosphoproteomics profiling reveals mTOR-signaling pathway as potentially altered in relapse.**

(A) Kinase activity in relapse ($n = 20$) versus non-relapse patients ($n = 30$), inferred using RoKAI. Bar length represent the RoKAI Activity score, and error bars indicate standard error. Color reflects the number of substrates for the kinase present in the data and used for kinase activity inference. Asterisks indicate statistical significance, where *** refers to FDR $q$-value < 0.001, **<0.01 and *<0.05. (B) Boxplot of protein and phospho-site intensities of two potential markers (AKT1S1:S183 and TBC1D4:S588), as function of disease relapse. $P$-values derived from a two-sample two-sided t-test statistical analysis, assuming equal variances. AKT1S1:S183 RFS Yes $n = 18$, AKT1S1:S183 RFS No $n = 26$, TBC1D4:S588 RFS Yes $n = 18$, TBC1D4:S588 RFS No $n = 24$, AKT1S1 and TBC1D4 RFS Yes $n = 20$, RFS No $n = 30$. Boxplot limits indicate the 25th and 75th percentiles as determined by R software; whiskers extend 1.5 times the interquartile range from the 25th and 75th percentiles, outliers are represented by dots. (C) Experimental design to evaluate the effect of 6 h mTOR inhibition with Torkinib in single spheroids ($n = 3$) at proteome and phosphoproteome level, assessed by LC-MSMS analysis. (D) Known phosphorylation sites substrates of mTOR levels in control (light orange) and 6 h with Torkinib (light green) conditions ($n = 3$ biological replicates). Height of the bar represents the mean intensity between replicates, and the error bar the standard error of the mean. (E) Gene Set Enrichment Analysis result for Hallmarks gene set "TNF-alpha signaling via NF-κB pathway", using proteomics ranked data (Torkinib treatment versus control). NES: normalized enrichment score, FDR q.val: FDR corrected $q$-value. (F) Protein intensity in control (light orange) and 6 h with Torkinib (light green) conditions corresponding to the top 5 proteins from the Hallmarks gene set "TNF-alpha signaling via NF-κB pathway" annotated in our data ($n = 3$ biological replicates). Height of the bar represents the mean intensity between replicates, and the error bar the standard error of the mean. (G) Known phosphorylation sites substrates of mTOR levels in wild type (dark green) and CAVIN1 KD (light green) conditions ($n = 3$ biological replicates). Statistical significance of the difference in abundance of each phosphorylation site was calculated using limma two-sample test, with FDR correction by Benjamini-Hochberg. Height of the bar represents the mean intensity between replicates, and the error bar the standard error of the mean.

CAVIN1-KD spheroids were smaller than their WT counterparts (Figs. 4C and EV4). Smaller size could indicate either lower growth rate or higher compaction of the spheroids. To identify the cause of the smaller size, we measured the spheroid cell viability to measure the spheroid cell density. As a result, we found that CAVIN1-KD spheroids had higher cell density than WT spheroids (Fig. 4D,E). This finding might reflect that CAVIN1 presence facilitates cell movement and detachment, as it happens during EMT transition and metastasis. Next, we repeated the spheroid formation assays in the CMS4-like CRC cell line SW480 (Berg et al, 2017). In contrast to HT29 cells, SW480 cells were not capable of forming spheroids in WT condition. However, most interestingly, we observed that CAVIN1-KD reverted partially this impairment (Fig. 4F).

Finally, we performed an invasion assay of WT and CAVIN1-KD spheroids in normal conditions and after EMT induction, respectively. This experiment revealed that, as expected, EMT-inducing cocktail increases invasiveness compared to WT cells, but most importantly, we observed that CAVIN1-KD abrogates the invasiveness potential of colorectal cancer cells (Fig. 4G,H; Appendix Fig. S8). Next, we extended the invasion assay using adherent-growing HT29 cell lines under the same conditions as before (WT, WT with EMT stimulation, and CAVIN1 KD), and additionally introduced EMT stimulation in CAVIN1 knockdown cells. We observed that CAVIN1 reduction blocked the increase in invasiveness typically induced by EMT stimulation (Fig. 4I). Finally, we employed mass spectrometry-based proteomics to profile the protein landscape of HT29 cells in response to CAVIN1 KD for 72 h (Fig. 4J). As expected, CAVIN1 KD cells showed a significant reduction in CAVIN1 levels, which, most importantly, was accompanied by a decrease in Vimentin (VIM) and other known markers of the mesenchymal phenotype, such as KIF5B and KILC1 (Moamer et al, 2019). Moreover, we observed an increase in epithelial markers, such as E-cadherin (CDH1) and Occludin (OCLN) (Lamouille et al, 2014). Altogether, our data suggest that CAVIN1 levels modulate the expression of genes related to the EMT transition, potentially leading to increased invasiveness.

## Hybrid-DIA phosphoproteomics profiling connect MTOR activation with better prognosis

To expand the knowledge of molecular mechanisms responsible for relapse in colorectal patients, we decided to interrogate the higher relapse-risk cohort further by performing phosphoproteomics profiling of the samples using a high-sensitivity hybrid-DIA analysis strategy (Martínez-Val et al, 2023). Hybrid-DIA is an intelligent MS acquisition strategy that we recently introduced to maximize the information retrieved from scarce biological samples, by combining targeted MS acquisition of well-known markers of signaling pathways with discovery-based phosphoproteomics. Less than 2 μg of peptide could be recovered from each colon cancer biopsy sample, which precludes conventional phosphoproteomics analysis. However, hybrid-DIA enabled us to profile ~1000 phosphorylation sites, and 35 markers of key signaling pathways extracted from the targeted analysis (Dataset EV4). Even though the phosphoproteomics coverage was restricted, we were able to use it to infer differential kinase activity as a function of relapse risk using RoKAI (Yılmaz et al, 2021). Interestingly, among the protein kinases down-regulated in patients that suffered relapse, there were several kinases involved in the mTOR signaling axis, including mTOR itself, but also p70S6K, AMPK1, and AKT1 (Fig. 5A). Specifically, we found two phosphorylation sites related to mTOR signaling, AKT1S1-S183 and TBC1D4-S588, among the pathway markers targeted in the hybrid-DIA analysis, that were statistically down-regulated in patients with colorectal cancer relapse after surgery (Fig. 5B). Most importantly, the corresponding protein abundance levels for AKT1S1 and TBC1D4 were not changing indicating that the observed downregulation in the phosphorylation sites is due to post-translational alteration of the signaling pathway and not simply due to changes in protein abundance (Fig. 5B). Based on these results, we decided to explore further the function of mTOR in colorectal cancer by measuring the proteome and phosphoproteome on CRC spheroids after in vitro inhibition of mTOR (Fig. 5C and Dataset EV5). Our phosphoproteomics data reflects that mTOR activity is abrogated after 6 h (Fig. 5D). Even though 6 h is a short time frame to induce significant alteration in the proteome, GSEA of the proteomes of spheroids treated with the mTOR inhibitor Torkinib versus those treated with DMSO showed significant upregulation of the TNF-alpha signaling via NF-κB pathway, which is a result of the increased levels of proteins such as FOSL1, JAG1, JUN, MAFF, and PLAUR (Fig. 5E,F). Interestingly, TNF-alpha signaling has been linked to epithelial to mesenchymal transition (Li et al, 2012; Zhang et al, 2014c; Hatanaka et al, 2021).

Therefore, we decided to investigate further if mTOR signaling in our colorectal cancer patient cohort in connection with CAVIN1

levels. Supporting our hypothesis, we observed that the Hallmarks gene set named "mTORC1 signaling" that includes the genes up-regulated through activation of the mTORC1 complex was significantly down-regulated in the EMT-like proteomics subtype (PROT subtype 1) (Fig. EV5A). Moreover, we also found that this signature is significantly down-regulated in patients diagnosed at stage III that relapse (Fig. EV5B). In contrast, we observed the opposite trend for CAVIN1, which was significantly over-expressed in the proteomics subtype 1 and in patients that suffered relapse (Fig. EV5C,D). This analysis was performed when analyzing all cohorts together, confirming that our previous findings regarding CAVIN1 relationship with relapse risk are translatable to new cohorts. Nevertheless, most importantly, these results suggest a likely relationship between CAVIN1, EMT mechanisms and mTOR regulation in colorectal cancer. To validate the potential connection between CAVIN1 and mTOR downregulation, we measured the phosphoproteomic profile of HT29 cells in response to CAVIN1 KD for 72 h (Dataset EV6). We found that several known mTOR substrates, including serine 2478 and serine 2481, were upregulated in response to the reduction in CAVIN1 levels, further confirming the potential link we previously described (Fig. 5G).

## Discussion

Here, we present a proteomics-centric discovery study on pre-metastatic colon cancer showing that tumor proteome profiles can stratify CRC patients into distinct molecular subtypes, which can be used to assist on therapy choice but also reflects the potential of proteomics to identify markers with prognostic potential. Using only proteomics-profiles, we can clearly stratify the patients within immune and stromal subtypes, with very good correlation with the classification obtained using CMScaller package and transcriptomics data (Eide et al, 2017a). The immune and stromal subtypes were also very well defined using the ESTIMATE (Yoshihara et al, 2013) scoring system with the proteomics data, which can greatly simplify the classification into these two subtypes. Additionally, the other tumor samples were classified into two different molecular subtypes, one with a metabolic signature and another with enrichment in canonical pathways for cancer. Although these two subtypes also match the molecular characteristics of the RNA-based CMS classification (CMS3 and CMS2, respectively) (Guinney et al, 2015), the correlation of patients assigned to these groups was not perfect. We identified a group of patients in our proteomics subtype 2 (metabolic enriched profile) who were classified as CMS3 or canonical by their transcriptomic profile. Supporting the proteomics-based subtyping, that group of patients were classified within the iCMS2 epithelial subtype defined by Joanito et al (Joanito et al, 2022). It could be of interest to investigate if there is an intermediate subtype, mixed with metabolic and canonical profiles that can further stratify colorectal cancer patients. Interestingly, previous works aiming to define a proteomics-based molecular classification of colorectal cancer did not identify such a metabolic subtype or CMS3-like subtype (Guinney et al, 2015). For instance, Vasaikar et al unified multi-omics subtypes, or UMS, eliminated the CMS3 subtype and assigned CMS3 tumors to other UMS subtypes (Vasaikar et al, 2019).

One important highlight of this study from a technical perspective is that, in contrast to previous works, we focused on

pre-metastatic cancer and our cohort is solely composed of tumor samples and not paired with healthy tissue, which better reflects the samples used in a clinical context for diagnostic and histopathology analysis. Moreover, we present a robust analysis pipeline that can control and correct for batch effects while preserving the biological information on the dataset. This strategy facilitates the possibility to classify tumor samples by their clinical and molecular traits, and not by technical characteristics. This opens up the potential of extending these cohorts with new additional samples, which mimics better the workflow in a clinical setup. Additionally, we have provided evidence on how a protein-centric strategy can increase the information obtained from transcriptional analysis, such as the differential regulation of the MRN complex at mRNA and protein level, which reflects that this multi-omics approach can identify post-translational regulatory mechanisms.

The main translational potential of this work was the identification of molecular mechanisms responsible for relapse in pre-metastatic colorectal patients. We did not find that any proteomics subtype showed statistically poorer prognosis than the others did when analyzed independently of the stage, not even in the EMT subtype (PROT subtype 1), which is supposed to phenotype found in the most aggressive tumors (Isella et al, 2015; Calon et al, 2015). However, we did find that the EMT signature was higher in TNM stage III tumors. Therefore, we explored protein biomarkers within the EMT subtype (PROT subtype 1) that are linked to higher risk of relapse. We found 38 proteins that were more abundant in EMT tumors and they were positively correlated with relapse. Among these potential candidates, we found three (PRELP, CAVIN1, and ITGA5), which also showed prognostic potential in the TCGA cohort. Altogether, our data seems to validate previous work that proposes that the tumor microenvironment and stroma is the main driver of good and poor prognosis in colorectal cancer (O'Shannessy et al, 2014; Calon et al, 2015) (Fig. 6A,B).

Although previous literature shows that elevated expression of a mesenchymal signature and poor prognosis in CRC samples is mainly due to tumor-associated stromal cells rather than by epithelial tumor cells (Calon et al, 2015; Isella et al, 2015), there is

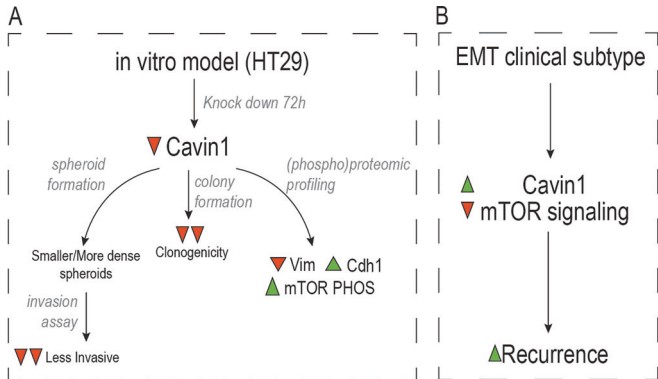

Figure 6. **Proposed model and validation for CAVIN1 link to colorectal cancer prognosis.**

(A) Recurrence hypothesis derived from proteomic profiling of colorectal cancer patients. (B) Conclusions derived from in vitro model of HT29 cell line with CAVIN knock down.

also evidence that CRC tumor cells can undergo EMT leading to higher invasiveness potential (Ieda et al, 2019). Therefore, we decided to evaluate CAVIN1 function not in the tumor micro-environment, but within the tumor cells, by using a 3D in vitro model of colorectal cancer. Interestingly, we observed that CAVIN1 plays a fundamental function in tumor cells, in relation to cell–cell interaction, EMT and invasiveness. This is consistent with previous studies linking CAVIN1 to metastatic potential in various tumor types, such as pancreatic cancer (Liu et al, 2014). Interestingly, CAVIN1 levels in cancer cells have also been shown to be associated with multidrug resistance (Yi et al, 2013). Therefore, future studies evaluating the involvement of CAVIN1 in drug resistance in colorectal cancer may provide further insights beyond our current findings.

Finally, we explored the potential of novel MS acquisition strategies to facilitate the analysis of the phosphoproteome from very low input tumor samples, such as the ones retrieved in our second cohort. Thanks to the combination of targeted MS analysis of key cellular signaling related phosphorylation sites with untargeted profiling of the global phosphoproteome, we could identify a kinase activity footprint in relapsed patients. Intriguingly, several kinases related to or within the mTOR signaling pathway were down-regulated in relapsed patients (Fig. 6A,B). We followed this hypothesis further and found out an anti-correlation at the proteome level between mTOR signaling and CAVIN1 levels. The results presented in here, although preliminary, opens up the possibility to investigate further the role of mTOR in colorectal cancer relapse risk. The strength of this study lies in the fact that hypotheses were derived directly from patient samples. However, validation was conducted solely in in vitro cell culture models, such as spheroids, which do not fully represent the observations made from patient biopsies. Further research to validate our findings must be carried out in in vivo models of this disease or in alternative cohorts. Moreover, evaluation on whether the cause of the relapse is due to remnant tumor cells not excised at the time of surgery or due to presence of metastasis that has not been yet diagnosed. If it is due to resistance to therapy, as it has been suggested in previous publications (Park et al, 2022; Lee et al, 2020), cannot be evaluated in this work.

In summary, we show that multi-omics analyses can be used to classify CRC tumors into fine-grained molecular subtypes and conclude that proteomics-based tumor classification has strong potential for identification of prognostic biomarkers in the most invasive and metastatic mesenchymal subtype, which can help identify risk of relapse for stage III patients.

# Methods

### Reagents and tools table

| Reagent/Resource | Reference or Source | Identifier or Catalog Number |
| --- | --- | --- |
| **Experimental models** | | |
| Tissue biopsies from primary colorectal cancers surgically resected (Homo Sapiens) | Biobank at the Department of Molecular Medicine, Aarhus University Hospital (Skejby, Denmark) | |
| HT-29 (Homo sapiens) | HT29 were kindly donated by Jesper B. Bramsen from Aarhus University Hospital. | HT-29 and SW480 cell line have been authenticated by STR profiling and cells have been kept in culture for no longer than 20 passages from the authenticated vial. |
| SW480 (Homo sapiens) | SW480 were kindly donated by Jesper B. Bramsen from Aarhus University Hospital. | HT-29 and SW480 cell line have been authenticated by STR profiling and cells have been kept in culture for no longer than 20 passages from the authenticated vial. |
| **Recombinant DNA** | | |
| **Antibodies** | | |
| **Oligonucleotides and other sequence-based reagents** | | |
| ON-TARGETplus Cavin1 siRNA SMARTpool | Dharmacon™ Reagents | L-012807-02-0005 |
| ON-TARGETplus Non-targeting Control siRNAs | Dharmacon™ Reagents | D-001810-01-20 |
| **Chemicals, enzymes, and other reagents** | | |
| Crystal violet solution, 1% Aqueous solution | Sigma-Aldrich | V5265 |
| Methanol | Sigma-Aldrich | 34860 |
| Opti-MEM, 500 mL | Thermo Scientific™ | 31985070 |
| Lipofectamine™ RNAiMAX Transfection Reagent, 1.5 mL | Thermo Scientific™ | 13778150 |
| SDS Solution, 20% Sodium Dodecyl Sulfate Solution, Molecular Biology/Electrophoresis, Fisher BioReagents™ | Fisher Scientific | BP1311-200 |
| Tris(2-carboxyethyl) phosphine hydrochloride | Sigma-Aldrich | C4706 |
| 2-Chloroacetamide, 98% | Sigma-Aldrich | C0267 |
| MagReSyn® Hydroxyl | ReSyn Biosciences | MR-HYX400 |
| Ethanol absolut ≥99.8%, AnalaR NORMAPUR® ACS, Reag. Ph. Eur. analyse reagens | VWR Chemicals | 20821.365 |
| Acetonitrile, ≥99.9% (GC), gradient grade, suitable for HPLC, Reag. Ph Eur, LiChrosolv® | Supelco | 1.00030 |
| 2-Propanol | Supelco | 107022 |
| Sequencing Grade Modified Trypsin | Promega | V5111 |
| Lysyl Endopeptidase® | Wako Chemicals | Mass Spectrometry Grade (Lys-C) |

| Reagent/Resource | Reference or Source | Identifier or Catalog Number |
|---|---|---|
| Trifluoroacetic acid | 8082601001 | Sigma-Aldrich |
| Formic Acid, LC-MS Grade | Thermo Scientific™ | 28905 |
| MagReSyn® Zr-IMAC HP | ReSyn Biosciences | MR-ZHP005 |
| Ammonia solution 25% | Supelco | 105432 |
| Sodium Fluoride | Sigma-Aldrich | 201154 |
| Sodium orthovanadate | Sigma-Aldrich | 567540 |
| Triethylammonium bicarbonate buffer, 1M, 500 mL | Supelco | 18597-500ML |
| Glycolic acid | Sigma-Aldrich | 124737-500G |
| MultiScreenHTS HV Filter Plate, 0.45 µm, clear, non-sterile | Millipore | MSHVN4550 |
| SureQuant™ Multipathway Phosphopeptide Standard | Thermo Scientific™ | A51745 |
| **Software** | | |
| Xcalibur | Thermo Scientific™ | tune version 1.1 or higher |
| Spectronaut | Biognosys | Version 16 |
| Perseus | https://maxquant.net/perseus/ | Version 1.6.15.0 |
| Skyline-daily | https://skyline.ms/project/home/begin.view | Version 21.1.9.353 |
| Moonshot App | https://github.com/thermofisherlsms/MoonshotApps | Version 1.3 |
| R | R foundation | Version 4.1.3 |
| Rstudio | Posit, PBC | R v4.1.3 |
| Graphpad | GraphPad Software, LLC | Version 9.5.0 for Windows |
| Leica Application Suite | Leica Microsystems | Version 4.12 |
| QuPath | Bankhead Group | 0.5.1 |
| STAR aligner | Gingeras Group | Version 1 |
| **Other** | | |
| Orbitrap Exploris™ 480 Mass Spectrometer | Thermo Scientific™ | BRE725539 |
| Orbitrap Astral Mass Spectrometer | Thermo Scientific™ | BRE725600 |
| Nanospray Flex™ Ion Source | Thermo Scientific™ | ES071 |
| FAIMS Pro™ Interface | Thermo Scientific™ | FMS02-10001 |
| Evosep One | Evosep | EV-1000 |
| EV1137 Performance Column – 30 SPD | Evosep | EV1137 |
| Evotip™ and Evotip Pure™ | Evosep | – |
| Reprosil-Pur C18 beads | Dr. Maisch, Ammerbuch, Germany | |

| Reagent/Resource | Reference or Source | Identifier or Catalog Number |
|---|---|---|
| Sonation column ovens for nano ESI | Berner Lab | PRSOV1 |
| IonOpticks Aurora™ column (15 cm–75 µm–C18 1.6 µm) | IonOpticks | 15 cm–75 µm–C18 1.6 µm |
| Butterfly Portfolio Heater | Phoenix S&T | PST-BPH |
| RNeasy Mini Kit ((250) | Qiagen NV | 74106 |
| High Sensitivity RNA ScreenTape | Agilent Technologies, Inc | 5067-5579 |
| Agilent 4200 TapeStation System | Agilent Technologies, Inc | G2991BA |
| KAPA RNA HyperPrep Kit with RiboErase (HMR) | F. Hoffmann-La Roche AG | KK8561 |
| xGen Dual Index UMI Adapter | Integrated DNA Technologies (IDT) | Product; TruSeq™–Compatible Duplex Y Adapter |
| NovaSeq 6000 Reagent Kits V1.0 (300 cycles; S1+S4) | Illumina, Inc | 20028313, 20012866 |
| NovaSeq 6000 Sequencing System | Illumina, Inc | 20012850 |
| Sep-Pak tC18 96-well Plate, 40 mg Sorbent per Well, 37–55 µm, 1/p | Waters Corporation | 186002320 |
| DMI3000 B Manual Inverted Microscope for Basic Life Science Research | Leica Microsystems | Discontinued |
| KingFisher Flex System | Thermo Scientific™ | |
| KingFisher™ plates for 96 deep-well format | Thermo Scientific™ | 95040450 |
| KingFisher™ plates for 96 standard and PCR formats | Thermo Scientific™ | 97002540 |
| KingFisher™ combitip for 96 deep-well format | Thermo Scientific™ | 97002570 |
| Corning® BioCoat® Matrigel® Invasion Chambers with 8.0 µm PET Membrane in two 24-well plates, 12/Pack, 24/Case | Corning Incorporated | 354480 |
| Forma 371 og 381 Steri Cycle CO2 inkubator, 184L | a/s Nino lab | Model 371 |
| Thermo Scientific™ Heraeus™ Multifuge™ | Thermo Scientific™ | X3/X3F Centrifuge Series |
| Hermle centrifuge | Ole Dich | Z216MK |
| Primers (for Pentaplex) | DNA Technology | custom |
| Clontech Titanium DNA Amplification kit | TAKARA BIO INC | 639240 |
| Sonicator (Sonic Dismembrator) | Thermo Scientific™ | Model 120 |

| Reagent/Resource | Reference or Source | Identifier or Catalog Number |
|---|---|---|
| Concentrator plus - Centrifuge Concentrator | Eppendorf | 5305000509 |
| NanoDrop™ 2000/ 2000c Spectrophotometers | Thermo Scientific™ | ND-2000 |

## Ethics declarations

Written informed consent was obtained previously from all patients and the study was conducted in accordance with local law and is approved by local institutional review boards and ethical committees. The study (MSmed: Mass spectrometric technology for next generation proteomics in systems medicine to Subtype and Discover Biomarkers of Colorectal cancer) was approved by the National Committee on Health Research Ethics (J. no. 1-10-72-80-17). The CRC biobank has been approved by the Danish Data Protection Agency (j. no. 1-16-02-27-10).

## Tissue samples collection

This study includes 412 tissue biopsies from primary colorectal cancers surgically resected (361 from the first cohort and 51 from the second cohort) from 371 patients in the period 2002–2020 (321 from the first cohort and 50 s cohort) were included in this study. The tissue biopsies were collected immediately after surgery, snap frozen in liquid nitrogen, and until used stored at −80 °C in the colorectal cancer biobank at the Department of Molecular Medicine, Aarhus University Hospital (Skejby, Denmark). All patients received standard of care treatment and follow-up in agreement with the guidelines (https://dccg.dk/retningslinjer/kolorektal-cancer/). The biopsies were collected fresh, from the surgical specimens resected as part of the patient's treatment for colorectal neoplasia, and immediately snap-frozen in liquid nitrogen and thereafter stored at −80 °C in the biobank. The fraction of cancer cells in collected biopsies was estimated by histological examination of hematoxylin and eosin stained sections, and when necessary, tumor biopsies were macroscopically trimmed to enrich the fraction of cancer cells. The cancer cell fractions of the finally included biopsies ranged between 40% and 90%. The MSI status of the samples was evaluated using a Pentaplex PCR. Samples with three or more markers showing satellite instability were classified as MSI, as recommended in Suraweera et al (Suraweera et al, 2002).

Post-operationally, tumors were histologically classified and staged according to the TNM staging system by pathological examination. Overall, included tumors were TNM stage II (52%) and stage III (46%) with similar representation of genders (45% female, 55% male) and a median age of 69 years at CRC diagnosis. Colon cancers and rectal cancers represented 83% and 17% of the tumors, respectively.

Researchers were blinded during the sample preparation and MS data acquisition. No sample size calculation was performed.

Written informed consent was obtained from all patients and the study was conducted in accordance with local law and is approved by local institutional review boards and ethical committees.

## Survival analysis

Patients were included in the relapse-free survival analysis if their tumor was micro-radically resected and if they were not diagnosed with another primary cancer, except non-melanoma skin cancer, within 36 months of primary CRC surgery. Relapse is defined as local relapse when it occurs at the same place as the original cancer (at the anastomosis, or in the tissue and lymph nodes in the region around the anastomosis) or as distant relapse if it occurs in organs or tissues far from the original cancer. Relapse-free survival was calculated as time between the date of primary CRC surgery and the date of relapse or death in connection to colorectal cancer (event) or end of follow-up (censoring). Relapse-free survival hazard ratio calculation was performed using Cox Proportional Hazards Ratio Model implemented in the function "coxph" in survival package (v3.4-0). For this analysis, when more than one biopsy was available for the same patient, proteomics data was averaged prior to hazard ratio calculation.

## Transcriptomic workflow

RNA sequencing of a total of 372 tissue samples was performed in three batches; The RNA sequencing of 147 batch 1 samples (Bramsen et al, 2017) and 34 batch 2 samples (Árnadóttir et al, 2018) is previously described. Batch 3 of 191 samples were subjected to total RNA sequencing as follows: Total RNA from 25 serial cryosections (10 μm) were extracted using the RNeasy Mini Kit (Qiagen), RNA integrity was assessed using the TapeStation RNA ScreenTape Assay (Agilent) and >98% of analyzed samples had a RNA integrity number (RIN) ≥ 6. RNA sequencing libraries were generated with the KAPA RNA HyperPrep Kit with RiboErase kit (Roche) using 500 ng total RNA as input. Sequencing libraries were sequenced on a NovaSeq Sequencing System (Illumina) at a maximum read length of 2*150 bp. and aiming for ~100 M read pairs per samples. Sequencing reads from all three batches were mapped to the human genome version GRCh38/hg38 using the STAR aligner (Dobin et al, 2013). Next, Transcripts per Million (TPM) values were calculated for GENCODE release 32 transcripts using Kalisto (Bray et al, 2016) and next summed for each gene. Quantile normalization and the ComBat function of the sva R-package (Leek et al, 2012) was applied to $\log_2(TPM + 1)$ expression values of protein-coding genes to reduce RNA sequencing batch effects. Consensus molecular subtypes (CMS) subtypes were called using the R-package CMScaller (Eide et al, 2017b) using batch-corrected $\log_2(TPM + 1)$ expression values for protein-coding genes as input.

## Sample preparation for proteomics analysis

Frozen tumors were cut individually into 25 slices of 20 μm, and lysed by adding 50 μl of boiling lysis buffer (6 M guanidinium chloride, 5 mM tris(2-carboxyethyl) phosphine (TCEP), 10 mM chloroacetamide (CAA), 100 mM Tris, pH 8.5, 1 mM Sodium Fluoride, 1 mM beta-glycerol phosphate, 1 mM sodium orthovanadate) and heating at 95 degrees for 10 min. The tumor lysate was sonicated for 20 s at 50% amplitude by micro tip probe sonication (Vibra-Cell VCX130, Sonics, Newtown, CT). Tumors from batches 1 to 3 were digested in solution. Lysates were directly diluted to 2 M GndCl before addition of proteases (Lys-C, Trypsin) and incubated

overnight at 37 °C. Protease activity was quenched by acidification with trifluoroacetic acid (TFA) to a final concentration of ~1%, and the resulting peptide mixture was concentrated using reversed-phase Sep-Pak C18 Cartridge (Waters). Peptides were eluted off the Sep-Pak with 150 μL 40% acetonitrile (ACN) followed by 150 μL 60% ACN. The ACN was removed by vacuum centrifugation for 40 min at 60 °C and the final peptide concentration was estimated by measuring absorbance at 280 nm on a NanoDrop (NanoDrop 2000C, Thermo Scientific).

Tumors from batches 4 to 9 were digested using the Protein Aggregation Capture (Batth et al, 2019) protocol. Protein extract was diluted with 100 mM Tris to 3 M guanidinium chloride and resuspended with acetonitrile to a final 70% concentration. 50 μL of MagReSyn® Hydroxyl beads were added to protein lysates and aggregation was performed in two steps of 1 min mixing at 1000 rpm, followed by 10 min pause each. Beads were subsequently washed three times with 1 mL 95% ACN and two times with 1 ml 70% EtOH. 300 μl of digestion buffer (50 mM Ammonium Bicarbonate) and proteases were added in the following proportions: trypsin 1:250 (enzyme:protein) and lysC 1:500 (enzyme:protein). Digestion was carried out overnight at 37 °C with looping mixing. Protease activity was quenched by acidification with trifluoroacetic acid (TFA) to a final concentration of 1%. 20 μL of peptide solution was taken for proteome analysis and divided into three Evotips.

Samples from the second cohort were subject to phosphoproteomics analysis. Peptides were concentrated on Sep-Pak (C18 Classic Cartridge, Waters, Milford, MA) and resuspended in 200 μL of 80% ACN, 5% TFA and 0.1 M Glycolic acid. Two rounds of phospho-enrichment were performed in the King-fisher robot using 5 μL of MagReSyn® Zr-IMAC HP beads (20 mg/ml). Enriched phosphopeptides were acidified with 10% TFA until pH < 3 and filtered to remove in-suspension particles (1 min, 500 g, MultiScreenHTS HV Filter Plate, 0.45 μm, clear, non-sterile). 0.5 μL of the SureQuant™ Multipathway Phosphopeptide Standard (100 fmol/μL) was added to each sample prior loading into Evotips for subsequent MS analysis.

## LC-MS/MS analysis

Samples for proteomics were analyzed in replicates on the Evosep One LC system using in-house packed 15 cm, 150 μm i.d. capillary column with 1.9 μm Reprosil-Pur C18 beads (Dr. Maisch, Ammerbuch, Germany) using the pre-programmed gradient for 60 samples per day. The column temperature was maintained at 60 °C using an integrated column oven (PRSO-V1, Sonation, Biberach, Germany) and interfaced online with the Orbitrap Exploris 480 MS (Thermo Fisher Scientific, Bremen, Germany) equipped with FAIMS and using Xcalibur (tune version 1.1 or higher). Spray voltage was set to 2.3 kV, funnel RF level at 40, and heated capillary temperature at 275 °C. Full MS resolutions were set to 120,000 at $m/z$ 200 and full MS AGC target was 300% with an IT of 45 ms. Mass range was set to 350–1400. AGC target value for fragment spectra was set at 1000%. 49 windows of 13.7 $m/z$ scanning from 361 to 1033 $m/z$ were used with an overlap of 1 Da. Resolution was set to 15,000 and IT to 22 ms and normalized collision energy was 27%. FAIMS compensation voltage was set to −45.

Phospho-proteomics data was acquired using the hybrid-DIA acquisition strategy with the Moonshot-app enabled through Thermo iAPI in an Exploris 480 (Martínez-Val et al, 2023). Samples were analyzed using an IonOpticks Aurora™ column (15 cm–75 μm–C18 1.6 μm) interfaced with the Orbitrap Exploris 480 MS using a Nanospray Flex™ Ion Source with an integrated column oven (PRSO-V1, Sonation, Biberach, Germany) to maintain the temperature to 50 °C. In all samples, spray voltage was set to 1.8 kV, funnel RF level at 40, and heated capillary temperature at 275 °C. Samples were acquired using the pre-programmed gradient for 20 samples per day. Hybrid-DIA data acquisition was performed using the moonshot API. For hybrid-DIA analysis, full MS resolution was set to 120,000 at $m/z$ 200 and full MS AGC target was 300% with an IT of 45 ms. Mass range was set to 350–1400. AGC target value for DIA scans was set at 1000%. Resolution was set to 30,000 and IT to 54 ms and normalized collision energy was 27%. DIA windows scanning from 472 to 1143 $m/z$ with 1 $m/z$ overlap were used (11 windows of 61.1 Da for 1 s cycle time at 30 K resolution). To enable non-isochronous injection times for MSx scans, the option must be enabled in Tune (available in Diagnostics > Method Setup). For hybrid-DIA inclusion lists, the retention time schedule was calculated from Survey Scans runs, where an inclusion list containing the $m/z$ and charge of the spiked-in IS peptides was used to specifically trigger their acquisition. Hybrid-DIA Moonshot API parameters were set up as follows: AGC target of 1e6, 116 ms injection time, NCE 27%, 1.5 Da isolation width, MS Trigger Intensity Threshold 1e5, 5 s of dynamic exclusion and 10 ppm of mass error.

## Proteomics data analysis

Raw files from proteomics data were searched using directDIA in Spectronaut (v16) using a human database (Uniprot reference proteome 2022 release, 20,598 entries) supplemented with common contaminants (246 entries). Carbamidomethylation of cysteine was set as a fixed modification, whereas oxidation of methionine and acetylation of protein N-termini were set as possible variable modifications. Cross-run normalization was turned off. Table with PG.Quantities was exported from Spectronaut into Perseus (v1.6.15.0). Samples from the cohort that did not meet the staging criteria and/or have less than 2000 protein groups quantified were removed from further analysis. Data was filtered to keep only protein groups quantified in at least 75% of the samples. Afterwards, contaminant and protein groups without gene names were removed from further analysis. Data was log2 transformed and missing values were imputed using a left censored distribution (1.8 downshift, 0.3 width) separately for each sample. Furthermore, all data analysis was performed in RStudio (R v4.1.3). Data was normalized using the "normalizecyclicloess" function from the limma package (v3.50.3). Batch correction due to digestion method used and processing batch was corrected using ComBat (Johnson et al, 2007). Technical replicates (MS run replicates) were merged by averaging the values.

Proteomics data was row-based scaled by median to calculate molecular subtypes using ConsensusClusterPlus (v1.58.0) R package, using a maximum of 6 clusters, "hc" as clustering algorithm and "Pearson" distance. Based on the results, $n = 4$ was chosen as the number of sample subtypes.

Row-wise median scaled proteomics data was annotated for stromal and immune score, as well as tumor purity using the estimate package (v1.0.13) (Yoshihara et al, 2013). For plotting purposes, stromal and immune scores were scaled across samples. Also, proteome signatures were collapsed into Hallmark gene sets using ssGSEA at GenePattern (Reich et al, 2006). Differential hallmark gene sets were selected using ANOVA and a *p*-value < 0.01.

Log2 proteomics data was used for differential expression analysis across subtypes using ANOVA test with a *q*-value (Benjamini-Hochberg) threshold <1e−9. Differentially regulated proteins between groups were used for Gene Ontology annotation using the "goana" function from the limma package.

Identification of potential recurrence markers was performed by fitting a linear model, using "lm" function from stats R package (v4.1.3), with protein levels of EMT markers (previously defined in the ANOVA differential expression analysis) and the metastasis event in the patients classified in the proteomics EMT subtype (subtype PROT 1). If more than one biopsy per patient was included in this subgroup, the protein abundance value was averaged.

The effect of time of EMT markers protein levels and the time of recurrence in patients for PROT subtype 1 was performed by fitting a Proportional Hazards Regression model using the "coxph" function from survival R package (v3.4-0).

## Phospho-proteomics data analysis (2nd COHORT)

From the hybrid-DIA runs, DIA scans were extracted using the HTRMS converter tool from Spectronaut (v15.4) indicating hybrid-DIA conversion in Conversion type. HTRMS resulting files were further used for directDIA search in Spectronaut (v18) using a human database (Uniprot reference proteome 2022 release, 20,598 entries) supplemented with common contaminants (246 entries). Carbamidomethylation of cysteine was set as a fixed modification, whereas oxidation of methionine, acetylation of protein N-termini and phosphorylation of serine, threonine and tyrosine were set as possible variable modifications. We filtered out 'b-ions' to prevent quantitative interference from heavy peptides. The maximum number of variable modifications per peptide was limited to 3. PTM localization Filter was checked and PTM localization cutoff was set to 0.75. Cross-run normalization was turned off. Phosphopeptide quantification data was exported and collapsed to site information using the Peptide Collapse (v1.4.2) plugin described in Bekker-Jensen et al (Bekker-Jensen et al, 2020a) and selecting "summing" as aggregation method in Perseus (v1.6.5.0). Data was log2 transformed and filtered to contain only phosphorylation sites quantified in at least 70% of the samples. Next, data was normalized using the "normalizecyclicloess" function from the limma package (v3.50.3) in R (v4.1.3). Missing values were imputed using a left censored distribution (1.8 downshift, 0.3 width) separately for each sample.

On the other hand, hybrid-DIA runs were also processed to extract the IS/ENDO multiplexed scans for targeted analysis. Multiplexed scans containing the internal standard and the endogenous peptide were extracted in an mzML file using the python GUI described in Martinez-Val et al (Martínez-Val et al, 2023). Resulting mzML files were loaded into a Skyline-daily (v21.1.9.353) to extract the intensity information of IS/ENDO

scans. The obtained data was used for injection time correction and peak area (AUC) calculation using the R-based (v4.0) as described in Martinez-Val et al (Martínez-Val et al, 2023). Intensities from ENDO peptides were normalized based on the IS peptide intensities, followed by a second normalization step to correct for loading bias, by using median intensity from DIA scans. For kinase activity inference, data from the global phosphoproteomics profiling was merged with the data from the targeted profiling, giving priority to the quantification obtained by targeted measurements if sites were identified by both strategies. Relative ratios relapse vs no-relapse were used as an input for RoKAI (Yılmaz et al, 2021) (v2.3.0).

## Multicellular colorectal cancer spheroid cell model

HT-29 and SW480 cell lines have been authenticated by STR profiling and cells have been kept in culture for no longer than 20 passages from the authenticated vial.

Cells were cultured in DMEM (Gibco, Invitrogen), supplemented with 10% fetal bovine serum (FBS, Gibco), 100 U/ml penicillin (Invitrogen), 100 µg/ml streptomycin (Invitrogen), at 37 °C, in a humidified incubator with 5% $CO_2$. Cultures were grown up to standard confluences of 70–90%.

Prior to spheroid formation HT-29 cells were seeded to P15 plates and grown to a confluence of 70–90%. Subsequently, cells were washed with Phosphate-buffered saline (PBS) (Gibco, Life Technologies) and detached from the plate with Trypsin. Following this, the cells were counted using a Corning® Cell Counter (Sigma-Aldrich). For the multicellular spheroid generation 7000 cells were seeded on ultra-low attachment 96-well plates (Corning CoStar, Merck). The spheroids were cultured for 96 h at 37 °C, in a humidified incubator with 5% $CO_2$. Cell medium was refreshed after 48 h, by aspirating half the old medium (making sure not to alter the spheroid) and adding the same amount of fresh medium. To achieve a knockdown, 1.25 µl of siRNA (Dharmacon) was added to 125 µl of opti-MEM (Gibco, Invitrogen). Additionally, 7.5 µl of Lipofectamine™ RNAiMAX Transfection Reagent (Thermo Fisher Scientific) was added to 125 µl of opti-MEM. Both solutions were added together and incubated for 3 min and added to cells grown with antibiotic free medium in a 6-well plate. After 24 h the cells were harvested and seeded for spheroid formation.

After 96 h, pictures of the spheroids were taken with a Leica DMI3000 B inverted microscope at 10x zoom + 5x zoom. The circumference of the spheroids was determined with the Invasion SpheroID ImageJ Analysis INSIDIA open-source macro running on ImageJ (Moriconi et al, 2017). Subsequently, the spheroids were collected by aspirating the DMEM medium followed by 2 PBS washes. To disaggregate the spheroids, and prepare for cell counting, the spheroids were treated with Accumax (Sigma-Aldrich) for 30 min at RT shaking at 750 rpm. The Accumax was aspirated, and the cells were resuspended in 90 µl of DMEM medium and 10 µl of CCK-8 solution (Tebu-bio) to measure the bio-reduction in the presence of an electron carrier, and hereby the cell number. The cells were incubated at 37 °C, in a humidified incubator with 5% $CO_2$ for 1 h. Consequently, the absorbance was measured at 450 nm with a FLUOstart Omega (BMGLabtech). The absorbance/relative cell number was divided by the circumference of the spheroids to determine the relative density of the spheroids. The data was both analyzed, and the figure created using Graphpad

Prism (version 9.5.0 for Windows, GraphPad Software, San Diego, California, USA, www.graphpad.com).

## In vitro invasion assay

The invasion of HT-29 cells was measured using the Matrigel 24-well plate 8.0-micron invasion chambers (Corning). The manufacturer's instructions were followed with the exception of the cell seeding, using a single spheroid per condition/replicate instead of a monolayer of cells. After seeding the spheroids on the matrigel membrane the plates were incubated prior to following steps. Spheroid staining of the invaded cells was performed with a crystal violet staining protocol whereby the membranes are placed on ice and washed twice 2X with cold PBS, followed by fixation for 10 min with ice-cold 100% methanol. Subsequently. The membranes were immersed in a solution of 0.5% crystal violet solution in 25% methanol for 10 min. Finally, they were washed with PBS and placed on microscope slides for visualization. The slides were visualized using a Leica DMI 6000B microscope at $10\times + 20\times$ magnification in grayscale. One-way ANOVA followed by Dunnett's multiple comparisons test was performed using GraphPad Prism version 10.2.0 (for Windows, GraphPad Software, Boston, Massachusetts, USA, www.graphpad.com) for statistical comparison.

## Clonogenic assay methods

Prior to seeding on a 6-well plate, Cavin1 was knocked down in HT-29 cells. To achieve a knockdown, 1.25 µl of siRNA (Dharmacon) was added to 125 µl of opti-MEM (Gibco, Invitrogen). Additionally, 7.5 µl of Lipofectamine™ RNAiMAX Transfection Reagent (Thermo Fisher Scientific) was added to 125 µl of opti-MEM. Both solutions were added together and incubated for 3 min and added to cells grown with antibiotic free medium in a 6-well plate. After 24 h the cells were trypsinised and diluted to 800 cells per 2 mL. To a 6-well plate 2 mL of cells was added, and incubated for 24 h. After 24 h, old medium was aspirated and fresh medium added. Consequently, the medium was replaced every 72 h. After 14 days, the colonies were stained using the Kwif-diff staining set (Fisher-Scientific). Consequently, counting of the colonies was done using the J2SE Clono-counter (Niyazi et al, 2007). One-way ANOVA followed by Dunnett's multiple comparisons test was performed using GraphPad Prism version 10.2.0 for statistical comparison.

## mTOR inhibition proteome and phosphoproteome profiling by LC-MSMS

mTOR inhibition in spheroids was achieved by treating HT-29 single spheroids at 4.5 µM of Torkinib (stock concentration: 10 mM) (previously identified HT-29 IC50). After 96 h spheroids were harvested by transferring single spheroids to a 1.5 ml tube and washing twice with ice-cold PBS followed by addition of 30 µl of boiling lysis buffer (5% sodium dodecyl sulfate (SDS), 5 mM tris(2-carboxyethyl) phosphine (TCEP), 10 mM chloroacetamide (CAA), 100 mM Tris, pH 8.5) and boiling at 95 °C for 10 min (shaking). The protein lysate was digested using the Protein Aggregation Capture protocol in the KingFisher robot. Briefly, the lysate was resuspended with acetonitrile to a final 70% concentration.

MagReSyn® Hydroxyl beads were added in a proportion 1:5 (protein:beads). Protein aggregation was performed in two steps of 1 min mixing at 1000 rpm, followed by 10 min pause each. Beads were subsequently washed three times with 1 ml 95% ACN and two times with 1 ml 70% EtOH. 100 µL of digestion buffer (50 mM Ammonium Bicarbonate) and proteases were added in the following proportions: trypsin 1:250 (enzyme:protein) and lysC 1:500 (enzyme:protein). Digestion was carried out overnight at 37 °C with looping mixing. Protease activity was quenched by acidification with trifluoroacetic acid (TFA) to a final concentration of 1%.

Samples were concentrated in a Speed-vac to a volume <20 µl and mixed with 200 µL loading buffer (80% ACN, 5% TFA, 0.1 M glycolic acid) and transferred to a KingFisher 96 deep-well plate. Additional KingFisher plates were prepared containing 500 µL of loading buffer, 500 µL of washing buffer 2 (80% ACN, 1% TFA) or 500 µL of washing buffer 3 (10% ACN, 0.2% TFA) each. For each sample, 5 µL of Zr-IMAC-HP beads (20 mg/mL) were suspended in 500 µL 100% ACN previously added to the KingFisher plates. For elution, 200 µL of elution buffer (1% $NH_4OH$) were prepared and transferred to KingFisher plates. Beads were washed in loading buffer for 5 min at 1000 rpm, incubated with the samples for 20 min with mixing at medium speed and subsequently washed in loading buffer, washing buffer 2 and washing buffer 3 for 2 min with mixing at fast speed. Phosphopeptides were eluted from the beads by mixing with elution buffer for 10 min at fast speed. Beads were collected back into the acetonitrile plate and the enrichment was repeated using the flow-through from the first enrichment and eluted in the same elution buffer as the first enrichment. Phosphopeptides were acidified with 30 µl of 10% TFA and loaded directly into Evotips for subsequent LC-MSMS analysis.

Proteome samples were analyzed using the EV1137 Evosep column (15 cm × 150 µm, 1.5 µm) and the column temperature was maintained at 40 °C using a butterfly heater (PST-ES-BPH-20, Phoenix S&T). Phosphoproteome samples were analyzed using the IonOpticks Aurora™ column (15 cm–75 µm–C18 1.6 µm) interfaced with the Orbitrap Exploris 480 MS using a Nanospray Flex™ Ion Source with an integrated column oven (PRSO-V1, Sonation, Biberach, Germany) to maintain the temperature to 50 °C. Spray voltage was set to 2.0 kV for proteome samples and 1.8 kV for phosphoproteome samples. For all samples, funnel RF level at 40, and heated capillary temperature at 275 °C. Samples were acquired using the pre-programmed gradient for 30 samples per day for the proteome and 20 samples per day for the phosphoproteome.

For proteome analysis, full MS resolution was set to 120,000 at $m/z$ 200 and full MS AGC target was 300% with an IT of 45 ms. Mass range was set to 350–1400. AGC target value for DIA scans was set at 1000%. Resolution was set to 15,000 and IT to 22 ms and normalized collision energy was 27%. 49 DIA windows of 13.7 Th scanning from 361 to 1033 $m/z$ with 1 $m/z$ overlap were used.

For phosphoproteome analysis, full MS resolution was set to 120,000 at $m/z$ 200 and full MS AGC target was 300% with an IT of 45 ms. Mass range was set to 350–1400. AGC target value for DIA scans was set at 1000%. Resolution was set to 30,000 and IT to 54 ms and normalized collision energy was 27%. 26 DIA windows of 25.9 Th scanning from 472 to 1143 $m/z$ with 1 $m/z$ overlap were used.

Raw files were searched using directDIA in Spectronaut (v17) using a human database (Uniprot reference proteome 2022 release,

20,598 entries) supplemented with common contaminants (246 entries). Carbamidomethylation of cysteine was set as a fixed modification, whereas oxidation of methionine and acetylation of protein N-termini were set as possible variable modifications. For phosphoproteomics analysis, phosphorylation of serine, threonine and tyrosine was included as well as variable modification, and PTM localization filter was turned on with a probability cutoff of 0.75. In all analysis, cross-run normalization was turned off.

## CAVIN1 knock-down proteomics and phosphoproteomic profiling by LC-MSMS

The invasion of HT-29 cells was measured using the Matrigel 24-well plate 8.0-micron invasion chambers (Corning). For the 2D invasion assay the manufacturer's instructions were followed. Staining of the cells was performed with a crystal violet staining protocol whereby the membranes are placed on ice and washed twice 2X with cold PBS, followed by fixation for 10 min with ice-cold 100% methanol. Subsequently. The membranes were immersed in a solution of 0.5% crystal violet solution in 25% methanol for 10 min. Finally, they were washed with PBS and placed on microscope slides for visualization. The slides were visualized using a Leica DMI 6000B microscope at $10\times + 20\times$ magnification in grayscale. The 3D invasion assay was done identically with the exception of the seeding of the cells. Here spheroids were grown for 96 h prior to seeding on the matrigel membrane, where after they were incubated following the manufacturers protocol. Analysis of the photos was done using the QuPath-0.5.1 software.

## Proteomics and phospho-proteomics data analysis of CAVIN1 knockdown HT29 cells

HT29 Cells were cultured in DMEM (Gibco, Invitrogen), supplemented with 10% fetal bovine serum (FBS, Gibco), 100 U/ml penicillin (Invitrogen), 100 µg/ml streptomycin (Invitrogen), at 37 °C, in a humidified incubator with 5% $CO_2$. Cultures were grown up to standard confluences of 70–90%. Subsequently, cells were washed with Phosphate-buffered saline (PBS) (Gibco, Life Technologies) and detached from the plate with Trypsin and transferred to a 6-well plate to a confluence of ~20%. To achieve a knockdown, 1.25 µl of siRNA (Dharmacon) was added to 125 µl of opti-MEM (Gibco, Invitrogen). Additionally, 7.5 µl of Lipofectamine™ RNAi-MAX Transfection Reagent (Thermo Fisher Scientific) was added to 125 µl of opti-MEM. Both solutions were added 25 together and incubated for 3 min and added to cells grown with antibiotic free medium in the 6-well plate. After 24 h the media was replaced with fresh DMEM supplemented with 10% FBS, and incubated for an additional 48 h. After the incubation time cells were harvested by and washing twice with ice-cold PBS followed by addition of 100 µl of boiling lysis buffer (5% sodium dodecyl sulfate (SDS), 5 mM tris(2-carboxyethyl) phosphine (TCEP), 10 mM chloroacetamide (CAA), 100 mM Tris, pH 8.5) and boiling at 95 °C for 10 min (shaking). The protein lysate was digested using the Protein Aggregation Capture protocol in the KingFisher robot. Briefly the lysate was resuspended with acetonitrile to a final 70% concentration. MagReSyn® Hydroxyl beads were added in a proportion 1:2 (protein:beads). Protein aggregation was performed in two steps of 1 min mixing at 1000 rpm, followed by 10 min pause each. Beads

were subsequently washed three times with 1 ml 95% ACN and two times with 1 ml 70% EtOH. 100 µL of digestion buffer (50 mM Ammonium Bicarbonate) and proteases were added in the following proportions: trypsin 1:250 (enzyme:protein) and lysC 1:500 (enzyme:protein). Digestion was carried out overnight at 37 °C with looping mixing. Protease activity was quenched by acidification with trifluoroacetic acid (TFA) to a final concentration of 1%. (This is almost the same protocol as the spheroids with the exception of the beads ratio).

For the proteome 750 ng of peptide was loaded directly into Evotips for subsequent LC-MSMS analysis. The remaining peptides were concentrated on Sep-Pak (C18 Classic Cartridge, Waters, Milford, MA) and resuspended in 200 µL of 80% ACN, 5% TFA and 0.1 M Glycolic acid. One round of phospho-enrichment was performed in the King-fisher robot using 10 µL of MagReSyn® Zr-IMAC HP beads (20 mg/ml). Enriched phosphopeptides were acidified with 10% TFA until pH <3. Following this, the phosphopeptides were loaded directly into Evotips for subsequent LC-MSMS analysis.

The samples were eluted online using an Evosep One system (Evosep Biosystems) and separated using an Evosep 8 cm (EV1109, Evosep) performance column connected to a steel emitter (EV1086, Evosep) and heated to 40 °C. Samples were analyzed using the pre-programmed gradient for 60 samples per day integrated with an Orbitrap Astral Mass Spectrometer (Thermo Fischer Scientific) applying 1800 V spray voltage, funnel radio frequency level at 40, and a heated capillary temperature set to 275 °C. The mass spectrometer was operated in positive mode. A full scan range of 380 to 980 $m/z$ were recorded in profile mode using a resolution of 240,000 at $m/z$, a normalized automatic gain control (AGC) target of 500%, and a maximum injection time of 3 ms. The fragment spectra were acquired in narrow-window data-independent acquisition (nDIA) (Guzman et al, 2024) mode, with a precursor mass range of 380 to 980 Da with 2 $m/z$ isolation windows without overlap, 299 scan events with a resolution of 24,000. Isolated precursors were fragmented in the HCD cell using 25% normalized collision energy, a normalized AGC target of 500%, and a maximum injection time of 2.5 ms. Phospho-proteomics data was acquired using with a 100SPD method. Most other settings being the same with the exception of the full scan range of 480 to 1080 $m/z$.

Raw files were searched using directDIA in Spectronaut (19.1.240724.62635) using a human database (Uniprot reference proteome 2023 release, 20,598 entries) supplemented with common contaminants (246 entries). Carbamidomethylation of cysteine was set as a fixed modification, whereas oxidation of methionine and acetylation of protein N-termini were set as possible variable modifications. For phosphoproteomics analysis, phosphorylation of serine, threonine and tyrosine was included as well as variable modification, and PTM localization filter was turned on with a probability cutoff of 0.75. In all analysis, cross-run normalization was turned off. Precursor information was exported from Spectronaut and collapsed to phosphorylation site using Perseus Peptide Collapse plugin (v.1.4.2) (Bekker-Jensen et al, 2020a). Resulting phosphorylation site table was then processed in R (v.4.3.1) for further analysis. For both protein and phosphorylation site tables the analysis was performed in parallel. Intensity was log2 transformed and normalized across samples using "normalizeCyclicLoess" function from limma (v3.58.1) package. Afterwards, data was filtered to removed identifications without

any valid quantification in one condition, and afterwards to contain at least 2 valid values in one of the assessed conditions. Next, missing values were imputed using "wrapper.impute.slsa" function from DAPAR (v1.34.5) package. Differentially regulated sites or proteins were identified using the moderated t-test implemented in limma (using method = "robust" and employing Benjamini-Hochberg method for multiple testing correction).

## Data availability

The mass spectrometry proteomics data have been deposited to the ProteomeXchange Consortium via the PRIDE partner repository. Colorectal cancer samples proteomics analysis data is available via ProteomeXchange with identifier PXD044246, in vitro validation experiments data is available with identifier PXD056331. The RNA sequencing data is available at https://genome.au.dk/library/GDK000008/. Both transcriptomics and proteomics data has been disaggregated for gender.

The source data of this paper are collected in the following database record: biostudies:S-SCDT-10_1038-S44320-025-00102-8.

## Peer review information

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

## Acknowledgements

Work at The Novo Nordisk Foundation Center for Protein Research (CPR) is funded in part by a donation from the Novo Nordisk Foundation (NNF14CC0001 and NNF24SA0098829). Part of this work has been funded as part of the MSmed project that has received funding from the European Union's Horizon 2020 Research and Innovation program under grant agreement no. 686547 and as part of EPIC-XS project under the grant agreement no. 823839. This project was supported by a generous grant from the Danish Agency of Higher Education and Science to establish the PLATO research infrastructure: Danish National Mass Spectrometry Platform for Proteomics and Biomolecular Imaging (grant no. 5229-00012B).

## Author contributions

**Ana Martinez-Val**: Conceptualization; Resources; Data curation; Software; Formal analysis; Supervision; Validation; Investigation; Visualization; Methodology; Writing—original draft; Project administration; Writing—review and editing. **Leander Van der Hoeven**: Data curation; Formal analysis; Validation; Investigation; Methodology; Writing—original draft; Writing—review and editing. **Dorte B Bekker-Jensen**: Conceptualization; Data curation; Formal analysis; Investigation; Methodology; Project administration; Writing—review and editing. **Margarita Melnikova Jørgensen**: Formal analysis; Investigation; Writing—review and editing. **Jesper Nors**: Resources; Formal analysis; Investigation; Writing—review and editing. **Giulia Franciosa**: Supervision; Validation; Investigation; Methodology; Project administration; Writing—review and editing. **Claus L Andersen**: Conceptualization; Resources; Supervision; Funding acquisition; Validation; Investigation; Writing—original draft; Project administration; Writing—review and editing. **Jesper B Bramsen**: Conceptualization; Resources; Data curation; Formal analysis; Supervision; Validation; Investigation; Methodology; Writing—original draft; Project administration; Writing—review and editing. **Jesper V Olsen**: Conceptualization; Resources; Supervision; Funding acquisition; Validation; Investigation; Visualization; Methodology; Writing—original draft; Project administration; Writing—review and editing.

Source data underlying figure panels in this paper may have individual authorship assigned. Where available, figure panel/source data authorship is listed in the following database record: biostudies:S-SCDT-10_1038-S44320-025-00102-8.

## Disclosure and competing interests statement

DBB-J is an employee at Evosep Biosystems. All other authors declare no competing interests.

# Expanded View Figures

**Figure EV1.  Proteomics analysis pipeline.**

(**A**) Frozen tumors were retrieved from the colorectal cancer biobank at the Department of Molecular Medicine, Aarhus University Hospital, Skejby, Denmark, processed using a cryotome for proteomics analysis. First three batches of samples were processed using in-solution digestion. Remaining batches were processed using Protein Aggregation Capture (PAC) protocol in the automatized King-Fisher platform. 10% of digested peptides was employed for two MS runs. Each sample was analyzed in duplicates in the Exploris480 coupled to EvosepOne using 60SPD pre-programmed gradient. MS runs were acquired in DIA mode using the FAIMS interface at CV- 45. MS datasets were acquired in three moments, batches 1 to 6 in October 2019, batches 7 and 8 in April 2020, and batch 9, comprising the 2nd cohort, was processed in July 2022. (**B**) Peptide Spectrum Matches (PSMs, in black) or peptides (in red) identified in quality control runs analysed before and in between the analysis of colorectal cancer protein batches. (**C**) Principal Component Analysis (PCA) showing MS run distribution based on protein intensity: (first row) using the output from Spectronaut, after data filtering for low quality runs, loess normalization and Left Censored Distribution imputation of missing values, (second row) after ComBat normalization to remove batch effect due to digestion procedure, (third row) after second ComBat normalization to remove batch effect due to sample batch proceeding. (**D**) PCA showing MS run distribution color coded by different clinical and demographic traits.

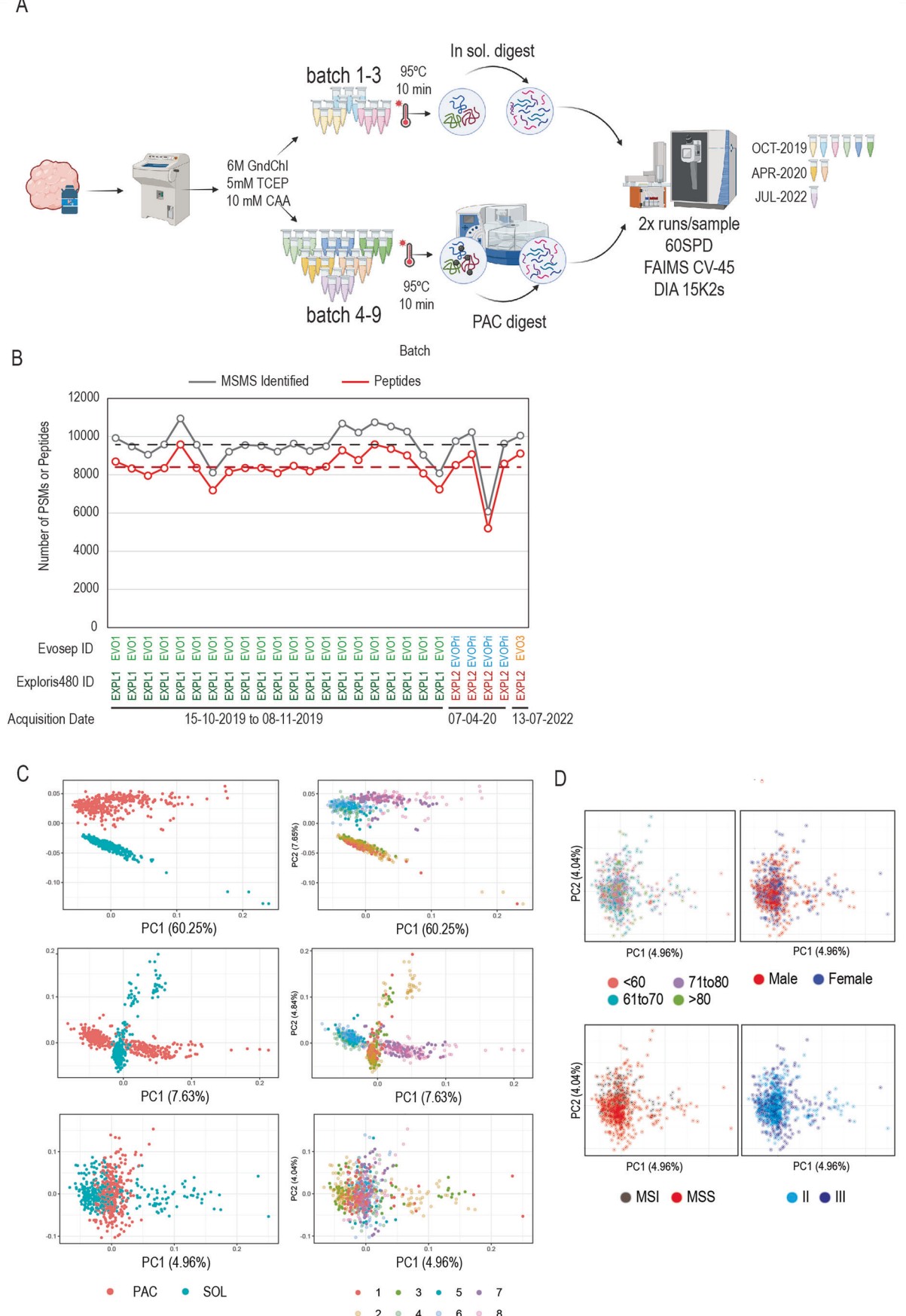

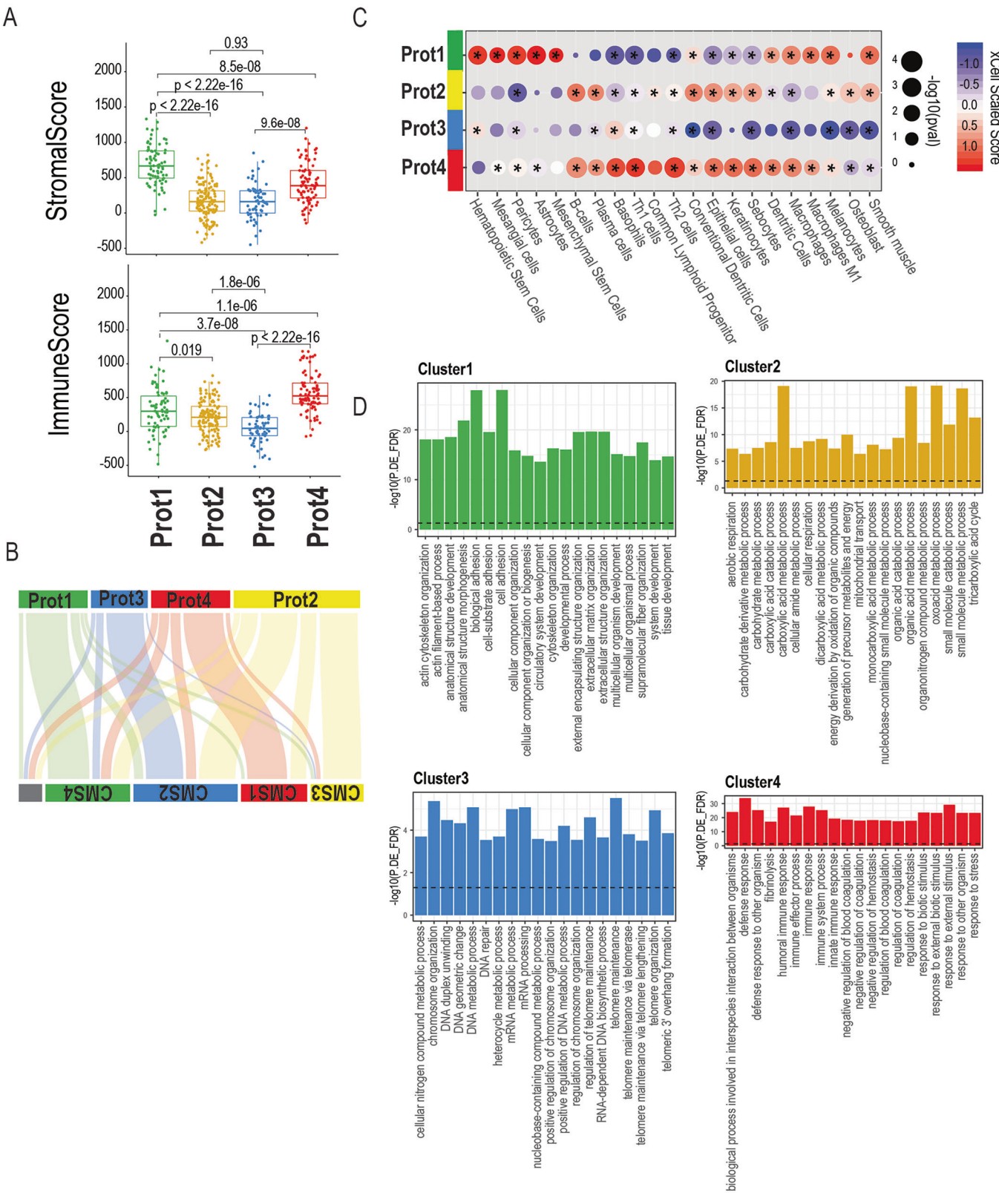

**Figure EV2. Differential regulation between proteomics subtypes.**

(A) Boxplots of Stromal and Immune score distribution across proteomics subtypes. Different subtypes are compared using a two-sample t-test, and the *p*-value of the test is plotted on top. Due to limitation of the binary digits accuracy in R, when *p*-value is close to zero, it is reported as $p < 2.22e{-}16$. Subtype 1 $n = 74$, subtype 2 $n = 136$, subtype 3 $n = 61$, subtype 4 $n = 85$. (B) Sankey plot showing the correspondence between proteomics subtypes and RNASeq-based CMS subtypes. Boxplot limits indicate the 25th and 75th percentiles as determined by R software; whiskers extend 1.5 times the interquartile range from the 25th and 75th percentiles, outliers are represented by dots. (C) Cell type enrichment results from xCell, using RNASeq data as input and grouped by proteomics subtype. Statistical significance is represented as *p*-value, reported using the statistical test developed for this purpose and built inside xCell tool (Aran et al, 2017). (D) Gene Ontology Overrepresentation Analysis (fisher test, one side, FDR BH) for terms enriched in each cluster. Bars represent the $-\log10$ FDR corrected *p*-value. Dashed line indicates FDR *p*-value of 0.05.

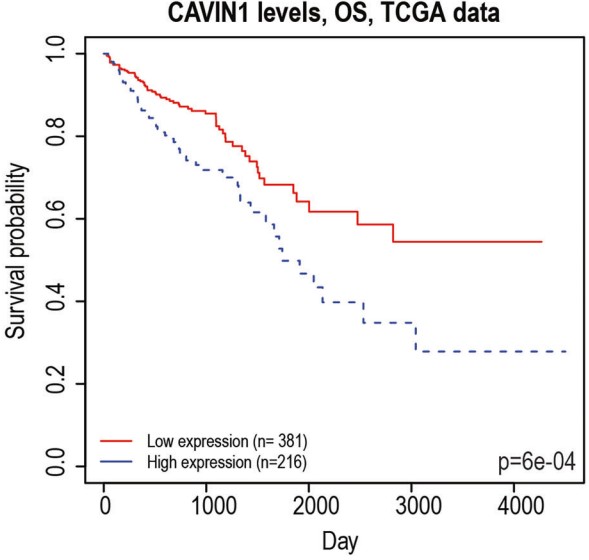

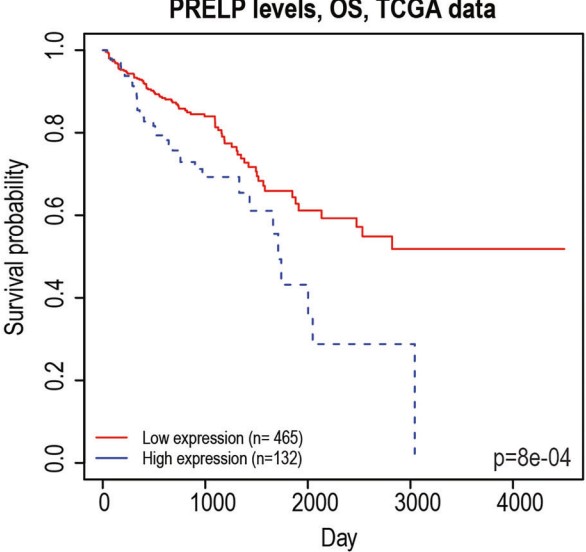

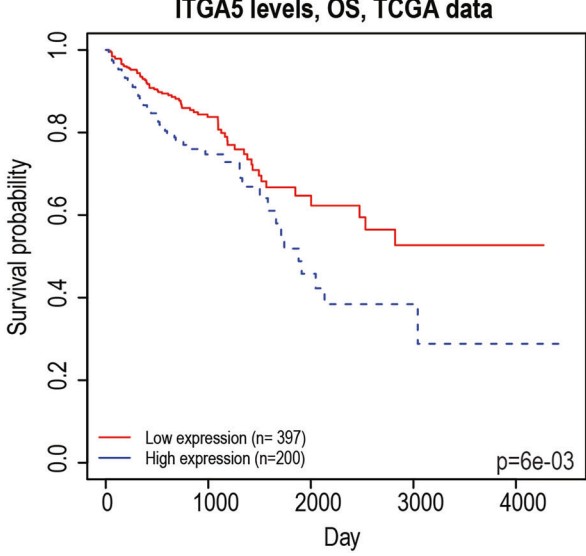

◀ **Figure EV3. Survival analysis of colorectal patients as a function of CAVIN1, PRELP and ITGA5 levels.**

Kaplan–Meier plots summarizing results from analysis of correlation between mRNA expression level and patient survival. Patients were divided based on level of expression into one of the two groups "low" (<mean expression of PROTEIN) or "high"(>= mean expression of PROTEIN). X-axis shows time for survival (years) and the y-axis shows the probability of survival, where 1.0 corresponds to 100 percent. Data was obtained from The Human Protein Atlas (Uhlen et al, 2017).

## Wildtype

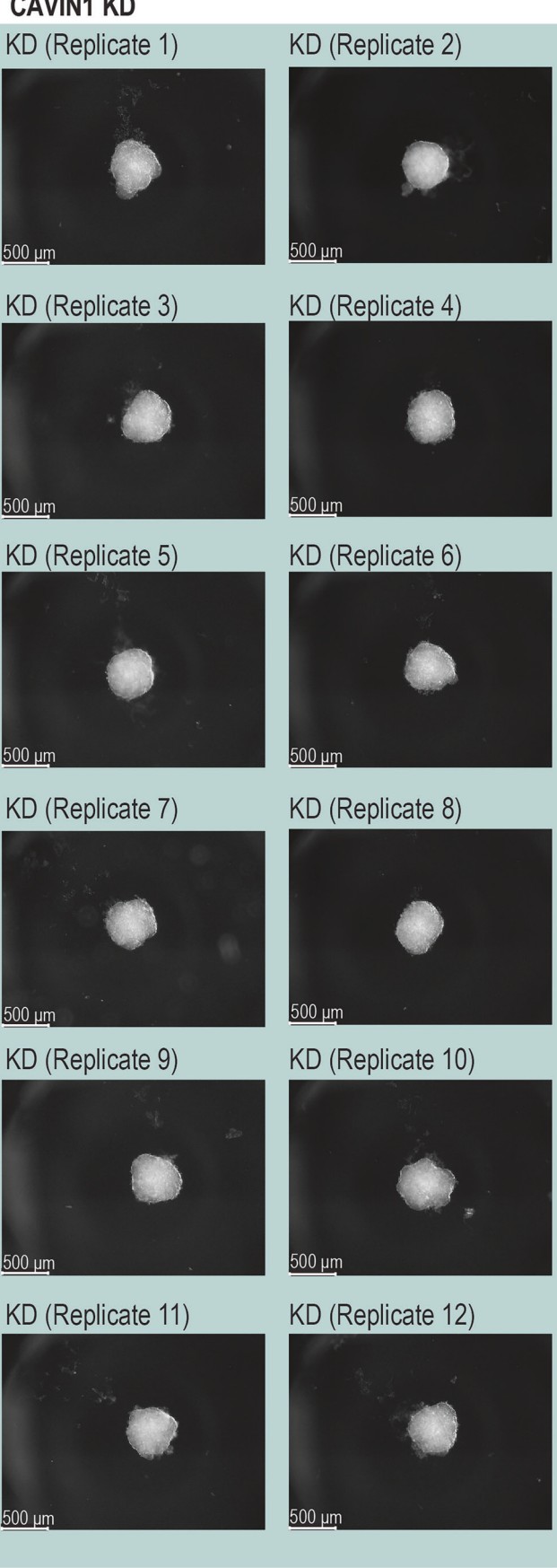

## CAVIN1 KD

◀ **Figure EV4. Spheroid formation assay in WT and CAVIN1-KD HT29 cells.**

Photographs of HT-29 derived spheroids, either wild type (left) or CAVIN1 knock down (right).

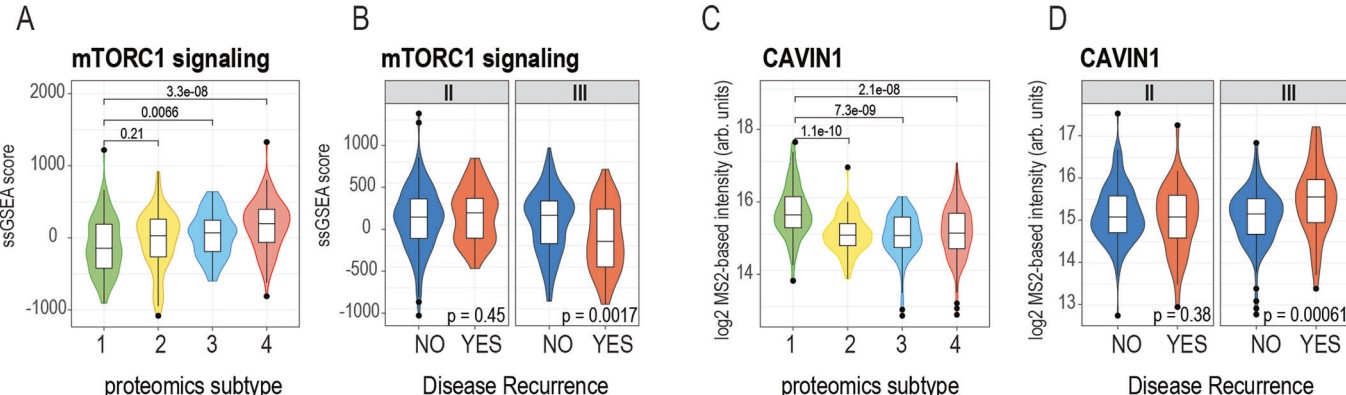

**Figure EV5.  Correlation between CAVIN1 levels and mTOR signaling with CRC proteomics subtypes.**

(A) Boxplot of the ssGSEA score for Hallmarks gene set "mTORC1 signaling" in each proteomic subtype (using 1st and 2nd cohort samples) with the corresponding significance value from a two-sample t-test between subtype 1 and the others. (B) Boxplot of the ssGSEA score for Hallmarks gene set "mTORC1 signaling" (using 1st and 2nd cohort samples) grouped based on tumor stage (II or III) and disease relapse. *P*-values derived from a two-sample two-sided t-test statistical analysis. (C) Boxplot of CAVIN1 protein levels in each proteomic subtype (using 1st and 2nd cohort samples) with the corresponding significance value from a two-sample t-test between subtype 1 and the others. (D) Boxplot of CAVIN1 protein levels (using 1st and 2nd cohort samples) grouped based on tumor stage (II or III) and disease relapse. Statistical *p*-values derived from a two-sample two-sided t-test analysis. For (A) and (C): subtype 1 $n = 97$, subtype 2 $n = 112$, subtype 3 $n = 72$, subtype 4 $n = 125$. For (B) and (D): Stage II RFS Yes $n = 42$, RFS No $n = 177$; Stage III RFS Yes $n = 54$, RFS No $n = 133$. For all panels, boxplot limits indicate the 25th and 75th percentiles as determined by R software; whiskers extend 1.5 times the interquartile range from the 25th and 75th percentiles, outliers are represented by dots.

