## [Peer Review File · Molecular Systems Biology]

Proteomics of colorectal tumors identifies the role of CAVIN1 in tumor relapse

Ana Martinez-Val, Leander Van der Hoeven, Dorte Bekker-Jensen, Margarita Jørgensen, Jesper Nors, Giulia Franciosa, Claus Andersen, Jesper Bramsen, and Jesper Olsen

Corresponding author(s): Jesper Olsen (Jesper.Olsen@cpr.ku.dk), Jesper Bramsen (bramsen@clin.au.dk), Claus Andersen (cla@clin.au.dk), Giulia Franciosa (giulia.franciosa@cpr.ku.dk), Ana Martinez-Val (ana.martinezdelval@cnic.es)

Review Timeline:

Submission Date:	10th Jan 25
Editorial Decision:	16th Jan 25
Revision Received:	28th Feb 25
Editorial Decision:	7th Mar 25
Revision Received:	28th Mar 25
Accepted:	2nd Apr 25

Editor: Jingyi Hou

Transaction Report:

This manuscript was transferred to Molecular Systems Biology following peer review at another journal.

16th Jan 2025

Manuscript Number: MSB-2025-12851P

Title: Proteomics of colorectal tumors identifies the role of CAVIN1 in tumor relapse

Author: Jesper Olsen

Dear Jesper,

Thank you for submitting your presubmission inquiry. I have now had the opportunity to review the revised manuscript and your point-by-point response. Overall, we believe the study has the potential to make an interesting contribution to the field.

Regarding the remaining comments from reviewers at the other journal, we consider the *in vivo* animal experiments to validate CAVIN1's role in tumorigenesis, as well as the further mechanistic investigation into how CAVIN1 modulates the mTOR pathway, to be beyond the scope of this study for publishing in *Molecular Systems Biology*. However, we suggest acknowledging these aspects as limitations and proposing them as potential directions for future research. Further, in response to the comments from Reviewers #1 and Reviewer #5 regarding the previous studies of CAVIN1 in tumor progression, we recommend more clearly highlighting the novelty and unique contributions of this study. Additionally, we think the value of proteome-based subtyping and the clinical relevance of the study could be emphasized more prominently. Lastly, the other concerns raised by Reviewer #5 should be carefully addressed in the revised manuscript. Please feel free to contact me in case you would like to discuss in further detail any of the issues raised by the reviewers.

On a more editorial level, please address the following issues :

- Please provide a .docx formatted version of the manuscript text (including legends for main figures, EV figures and tables). Please make sure that the changes are highlighted to be clearly visible.
- Please provide individual production quality figure files as .eps, .tif, .jpg (one file per figure).
- Please provide a .docx formatted letter INCLUDING the reviewers' reports and your detailed point-by-point responses to their comments. As part of the EMBO Press transparent editorial process, the point-by-point response is part of the Review Process File (RPF), which will be published alongside your paper.
- Please note that all corresponding authors are required to supply an ORCID ID for their name upon submission of a revised manuscript.
- We replaced Supplementary Information with Expanded View (EV) Figures and Tables that are collapsible/expandable online (see examples in <http://msb.embopress.org/content/11/6/812>). A maximum of 5 EV Figures can be typeset. EV Figures should be cited as 'Figure EV1, Figure EV2' etc... in the text and their respective legends should be included in the main text after the legends of regular figures. Additional Tables/Datasets should be labeled and referred to as Table EV1, Dataset EV1, etc. Legends have to be provided in a separate tab in case of .xls files. Alternatively, the legend can be supplied as a separate text file (README) and zipped together with the Table/Dataset file. For the figures and tables that you do NOT wish to display as Expanded View figures, they should be bundled together with their legends in a single PDF file called *Appendix*, which should start with a short Table of Content. Each legend should be below the corresponding Figure/Table in the Appendix. Appendix figures and tables should be referred to in the main text as: "Appendix Figure S1, Appendix Figure S2, Appendix Table S1" etc. See detailed instructions regarding expanded view here: <https://www.embopress.org/page/journal/17444292/authorguide#expandedview>.
- Before submitting your revision, primary datasets (and computer code, where appropriate) produced in this study need to be deposited in an appropriate public database (see <http://msb.embopress.org/authorguide-dataavailability> <https://www.embopress.org/page/journal/17444292/authorguide#dataavailability>). Please remember to provide a reviewer password if the datasets are not yet public. The accession numbers and database should be listed in a formal "Data Availability" section (placed after Materials & Method) that follows the model below (see also <https://www.embopress.org/page/journal/17444292/authorguide#dataavailability>). Please note that the Data Availability Section is restricted to new primary data that are part of this study.
Data availability
The datasets (and computer code) produced in this study are available in the following databases:
 - RNA-Seq data: Gene Expression Omnibus GSE46843 (<https://www.ncbi.nlm.nih.gov/geo/query/acc.cgi?acc=GSE46843>)
 - [data type]: [name of the resource] [accession number/identifier/doi] ([URL or identifiers.org/DATABASE:ACCESSION])*** Note - All links should resolve to a page where the data can be accessed. ***

-At EMBO Press we ask authors to provide source data for the main figures. Our source data coordinator will contact you to discuss which figure panels we would need source data for and will also provide you with helpful tips on how to upload and organize the files.

- Our journal encourages inclusion of *data citations in the reference list* to directly cite datasets that were re-used and obtained from public databases. Data citations in the article text are distinct from normal bibliographical citations and should directly link to the database records from which the data can be accessed. In the main text, data citations are formatted as follows: "Data ref: Smith et al, 2001". In the Reference list, data citations must be labeled with "[DATASET]". A data reference must provide the database name, accession number/identifiers and a resolvable link to the landing page from which the data can be accessed at the end of the reference. Further instructions are available at .

- We updated our journal's competing interests policy in January 2022 and request authors to consider both actual and perceived competing interests. Please review the policy <https://www.embopress.org/competing-interests> and update your competing interests if necessary. Please use the heading "Disclosure statement and competing interests".

- All Materials and Methods need to be described in the main text using our 'Structured Methods' format. According to this format, the Methods section includes a Reagents and Tools Table (listing key reagents, experimental models, software and relevant equipment and including their sources and relevant identifiers) followed by a Methods and Protocols section describing the methods, ideally using a step-by-step protocol format. The aim is to facilitate adoption of the methodologies across labs. Please download and fill our Reagents and Tools Table template (.docx), which you can find in our author guidelines: <https://www.embopress.org/page/journal/17444292/authorguide#structuredmethods>.

An example of a Method paper with Structured Methods can be found here: <https://www.embopress.org/doi/10.15252/msb.20178071>.

- Regarding data quantification:

Please ensure to specify the name of the statistical test used to generate error bars and P values, the number (n) of independent experiments (please specify technical or biological replicates) underlying each data point and the test used to calculate p-values in each figure legend. Discussion of statistical methodology can be reported in the materials and methods section, but figure legends should contain a basic description of n, P and the test applied.

Graphs must include a description of the bars and the error bars (s.d., s.e.m.).

- Please provide a "standfirst text" summarizing the study in one or two sentences (approximately 250 characters, including space), three to four "bullet points" highlighting the main findings and a "synopsis image" (550px width and 400-600 px height, PNG format) to highlight the paper on our homepage.

Here are a couple of examples:

<https://www.embopress.org/doi/10.15252/msb.20199356>

<https://www.embopress.org/doi/10.15252/msb.20209475>

<https://www.embopress.org/doi/10.15252/msb.209495>

When you resubmit your manuscript, please download our CHECKLIST (<https://www.embopress.org/pb-assets/embo-site/EMBO%20Press%20Author%20Checklist-1642513524327.xlsx>) and include the completed form in your submission.

Please note that the Author Checklist will be published alongside the paper as part of the transparent process (<https://www.embopress.org/page/journal/17444292/authorguide#transparentprocess>).

If you feel you can satisfactorily deal with these points and those listed by the referees, you may wish to submit a revised version of your manuscript. Please attach a covering letter giving details of the way in which you have handled each of the points raised by the referees. A revised manuscript may be once again subject to review and you probably understand that we can give you no guarantee at this stage that the eventual outcome will be favorable.

I look forward to receiving your revised manuscript soon.

Jingyi

Jingyi Hou, PhD
Senior Editor
Molecular Systems Biology

We realize that it is difficult to revise to a specific deadline. In the interest of protecting the conceptual advance provided by the work, we recommend a revision within 3 months (16th Apr 2025). Please discuss the revision progress ahead of this time with the editor if you require more time to complete the revisions. Use the link below to submit your revision:

IMPORTANT: When you send your revision, we will require the following items:

1. the manuscript text in LaTeX, RTF or MS Word format
2. a letter with a detailed description of the changes made in response to the referees. Please specify clearly the exact places in the text (pages and paragraphs) where each change has been made in response to each specific comment given
3. three to four 'bullet points' highlighting the main findings of your study
4. a short 'blurb' text summarizing in two sentences the study (max. 250 characters)
5. a 'thumbnail image' (550px width and max 400px height, Illustrator, PowerPoint or jpeg format), which can be used as 'visual title' for the synopsis section of your paper.
6. Please include an author contributions statement after the Acknowledgements section (see <https://www.embopress.org/page/journal/17444292/authorguide>)
7. Please complete the CHECKLIST available at (<https://bit.ly/EMBOPressAuthorChecklist>).

Please note that the Author Checklist will be published alongside the paper as part of the transparent process (<https://www.embopress.org/page/journal/17444292/authorguide#transparentprocess>).

See also figure legend guidelines: <https://www.embopress.org/page/journal/17444292/authorguide#figureformat>

9. Please note that corresponding authors are required to supply an ORCID ID for their name upon submission of a revised manuscript (EMBO Press signed a joint statement to encourage ORCID adoption).

(<https://www.embopress.org/page/journal/17444292/authorguide#editorialprocess>)

Currently, our records indicate that the ORCID for your account is 0000-0002-4747-4938.

Link Not Available

11. Include a Reagents and Tools Table as part of the Methods section, which can be downloaded from our author guidelines (<https://www.embopress.org/page/journal/17444292/authorguide#structuredmethods>)

*** PLEASE NOTE *** As part of the EMBO Press transparent editorial process initiative (see our Editorial at <https://dx.doi.org/10.1038/msb.2010.72>), Molecular Systems Biology publishes online a Review Process File with each accepted manuscripts. This file will be published in conjunction with your paper and will include the anonymous referee reports, your point-by-point response and all pertinent correspondence relating to the manuscript. If you do NOT want this File to be published, please inform the editorial office at msb@embo.org within 14 days upon receipt of the present letter.

Point-by-point rebuttal letter to REVIEWER COMMENTS

Manuscript Number: **MSB-2025-12851P**

“Proteomics of colorectal tumors identifies the role of CAVIN1 in tumor relapse”
by Martinez-Val et al.

Please, find below our responses to reviewers' requests and questions highlighted in blue.

Reviewer #1 (Remarks to the Author):

The authors have performed more experiments to address most of my concerns. The quality of manuscript is largely improved. However, it's suggested to perform additional in vivo experiments to validate the roles of CAVIN1 in tumor invasiveness. For example, nude mice were employed to analyze the invasive ability of CAVIN1 KD cells.

RESPONSE: We appreciate the positive feedback from this reviewer. Although we agree that performing additional in vivo experiments will be of great interest, we consider that such costly and laborious experiments are beyond the scope of the current work, as we have indicated in the Discussion of the manuscript (line 521 to 528), which reads:

“The strength of this study lies in the fact that hypotheses were derived directly from patient samples. However, validation was conducted solely in in vitro cell culture models, such as spheroids, which do not fully represent the observations made from patient biopsies. Further research to validate our findings must be carried out in in vivo models of this disease or in alternative cohorts. Moreover, evaluation on whether the cause of the relapse is due to remnant tumor cells not excised at the time of surgery or due to presence of metastasis that has not been yet diagnosed. If it is due to resistance to therapy, as it has been suggested in previous publications^{63,64}, cannot be evaluated in this work.”

Reviewer #3 (Remarks to the Author):

The authors have addressed all my questions. Thanks.

RESPONSE: We thank you.

Reviewer #5 (Remarks to the Author):

In response to Reviewer 4's concerns, the authors' answers need further clarification. They found that the PROT subtype and the biomarker CAVIN1 are linked to recurrence, but it's unclear whether they are associated with local recurrence or distant metastasis. Additionally, did the patients receive any adjuvant chemotherapy or radiation therapy after surgery? Are the PROT subtype and CAVIN1 related to the treatments the patients received, such as chemotherapy or targeted therapy?

RESPONSE: The primary treatment for all patients included in our cohort was curative intent surgery. Patients were treatment-naïve prior to surgery. Accordingly, any relapse event must mean that residual disease persisted in the patient after surgery. Moreover, in our original discovery cohort there was no local recurrence cases, therefore, in case of distant relapse, it can be argued that surgical removal of the primary tumor was complete, but that occult metastatic spreading (undetectable by diagnostic CT imaging) had already occurred at time of surgery, and that this spread later manifested as detectable lesions.

In relation to the reviewer's comment regarding response to treatment, we do acknowledge, that a minor fraction of patients, mainly patients diagnosed with stage III disease, received adjuvant chemotherapy after surgery, in accordance with Danish national guidelines. Specifically, there are different recommendations for adjuvant chemotherapy for stage III colon cancer, stage III rectal cancer, and stage II colon cancer and rectal cancer. Patients are typically treated with 5-fluorouracil (5-FU) as standard therapy, and in high-risk cases, oxaliplatin is added, unless the patient is over 70 years old, in which case they only receive 5-FU. Treatment durations range from 3 to 6 months, and patients generally do not receive treatment if they are over 75 years (for stage II) or 80 years (for stage III colon and rectal cancer). Moreover, dMMR patients typically do not respond to 5-FU and are therefore not treated.

Taking all this into consideration, for a patient relapsing despite both surgery and adjuvant therapy, the relapsing cells are likely resistant to adjuvant chemotherapy. However, we attempted to gather treatment information from patients in our cohort; but the data were insufficient to derive meaningful insights regarding the relapse risk associated with treatment resistance, and therefore to connect adjuvant treatment effect to CAVIN1 levels.

The authors used different digestion methods, resulting in multiple batches of data. Although they applied batch effect correction twice, it's important to analyze whether these different batch effects have any correlation with recurrence.

RESPONSE: We understand the concern of the reviewer regarding the experimental variability introduced due to different processing batches and digestion strategies. To ascertain that sample allocation to experimental batches did not introduce a bias into the analysis, we performed Cox Multivariate Analysis to evaluate whether any particular batch correlated with relapse in a significant manner (Figure A). As observed in Figure A below, none of the experimental batches (1 to 8) showed significant correlation with recurrence. As a reference, on Figure B, we have represented the sample distribution of patients with and without recurrence among the 8 batches analysed. In this regard, we also did not observe any batch effect on the CAVIN1 levels (Figure C).

Figure 1. (A) Cox multivariate analysis of relapse in the colorectal cancer patient experimental cohort to assess correlation between different traits: processing batch, gender, MSI status (0=MSS, 1=MSI) and Stage (0=II, 1=III). (B) Distribution of samples across batches, where the height of the bar indicates the total number of samples. Each bar is divided between number of patients that suffered relapsed (in black) and those that did not (grey). Top graph represents total number of samples and bottom graph represent the same data as percentages. (C) Distribution of measured CAVIN1 protein levels across the different experimental batches. Protein distribution is represented as a boxplot. Anova t-test indicates that distribution of CAVIN1 protein intensities is not different between batches.

Moreover, this study analyzed tissue samples from 361 CRC patients using proteomics and identified the role of CAVIN1 in colorectal cancer. However, the authors only conducted simple in vitro experiments to show that CAVIN1 promotes cell migration. More in vivo animal experiments are needed to provide further insight into CAVIN1's function in CRC. Previous research has already established a connection between CAVIN1 and tumor progression, so what is the novel contribution of this study?

On the mechanistic side, the authors have merely indicated an association between CAVIN1 and the mTOR pathway through omics approaches. Additional experiments, like Western Blotting and functional rescue assays, are necessary to confirm how CAVIN1 regulates the mTOR signaling pathway and influences cell migration. The specific mechanism by which CAVIN1 modulates the mTOR pathway has not yet been explained.

RESPONSE: We appreciate the suggestion and we acknowledge that further in vivo experiments would be very valuable to validate the hypothesis derived from our systematic evaluation of colorectal cancer signatures. Moreover, we are aware that CAVIN1 has been previously linked to tumour progression, as we specified in the current manuscript (lines 95 to 101).

Our work focuses on the use of proteomics to define signatures that allow to stratify colorectal cancer patients, and how we can shortlist potential biomarkers from such data, as CAVIN1 in this case. Our data reflects a link between CAVIN1 and EMT molecular phenotype in colorectal cancer by means of proteomics, which, to our knowledge have not previously been described. Finally, we consider that additional in vivo experiments to be out of scope of the present work but we have acknowledged such limitations in the Discussion of the revised paper (lines 521 to 525), which reads:

“The strength of this study lies in the fact that hypotheses were derived directly from patient samples. However, validation was conducted solely in in vitro cell culture models, such as spheroids, which do not fully represent the observations made from patient biopsies. Further research to validate our findings must be carried out in in vivo models of this disease or in alternative cohorts.”

While the authors included an additional cohort to show that CAVIN1 has higher recurrence rates in the EMT subtype, this cohort consists of only 50 cases, and they need to justify the validity of this sample size.

RESPONSE: We understand the concern regarding the sample size of our validation cohort, and we agree that the size might limit the interpretation of the results. We have acknowledged this limitation in the revised work. However, to strengthen the validation of our data, we also have performed cross-validation of our findings in additional dataset, such from Li, Wang, and colleagues (PMID: 38086380), which is shown in Supplementary Figure 5C and 5D.

Clinically, the authors claim that CAVIN1 is associated with tumor recurrence. However, previous proteomic studies which have identified many biomarkers in several independent cohorts related to tumor recurrence. What sets CAVIN1 apart from these other markers? Furthermore, the authors should confirm the expression of CAVIN1 associate with tumor recurrence using immunohistochemistry (IHC) across multiple cohorts.

Additionally, since this biomarker is only associated with recurrence in the EMT subtype, how can it be applied clinically?

RESPONSE: Previous studies reporting alternative biomarkers are highly valuable, as we do not consider that certain biomarkers need to be set apart from others, but, on the contrary, biomarkers need to be considered as complementary to each other to derive panels that best reflect the tumour recurrence in each patient. In this work, we focused on CAVIN1 since it has not previously been identified as a biomarker of Colorectal Cancer survival by ATCG data, but there was prior evidence of the relevance of this protein in tumour progression as previously stated by this reviewer. Altogether we decided to follow up of this protein, which can potentially complement the information from other publications in the field.

Although we consider essential to validate CAVIN1 using IHC, recruiting enough samples from multiple cohorts as suggest by this reviewer, is currently out of the scope of the present manuscript.

Regarding clinical application of CAVIN1 being limited to EMT subtype, our data shows that colorectal cancer is a highly heterogenous disease, where EMT subtype differs greatly from the other three molecular subtypes identified in our cohort. Therefore, we consider it important to identify specific subtype biomarkers, to evaluate the relapse risk as a function of its molecular subtype.

7th Mar 2025

Manuscript Number: MSB-2025-12851

Title: Proteomics of colorectal tumors identifies the role of CAVIN1 in tumor relapse

Author: Ana Martinez-Val

Leander Van der Hoeven

Dorte Bekker-Jensen

Margarita Jørgensen

Jesper Nors

Giulia Franciosa

Claus Andersen

Jesper Bramsen

Jesper Olsen

Dear Jesper,

Thank you for the submission of your revised manuscript to Molecular Systems Biology. We have now had the chance to go through the revised manuscript and your point-by-point response. Overall, we think the concerns of the reviewers at the other journal have been satisfactorily addressed, and the study is now suited for publication in Molecular Systems Biology. Before we can formally accept the study for publication, we would ask you to address the following editorial-level points:

1. Please provide email addresses for all corresponding authors in the title page of the manuscript.
2. Please remove the "Author Contribution" section from the manuscript.
3. Please note that figure panels need to be called out, and all callouts should be listed sequentially.
4. Data availability: please remove the reviewer access codes and provide specific URLs for the PXD056331 and PXD044246 datasets.
5. "Declaration of interests" should be renamed to "DISCLOSURE AND COMPETING INTERESTS STATEMENT".
6. Citations should be listed in alphabetical order.
7. Author checklist: Please enter the names of all corresponding authors. The fields in the last column (the pink one) should be blank if the response is "Not Applicable".
8. Appendix: The appendix file needs to be in PDF format. Please add page numbers of the listed items in the Table of Content.
9. "Ethics declaration" should be part of Methods.
10. Reagents and Tools table should be removed from the manuscript file and uploaded as a separate .docx file.
11. The synopsis image should be uploaded as a separate in .jpg or .png file that is exactly 550 pixels wide and 300-600 pixels high. Please remove both the synopsis text and image from the manuscript file.
12. Section order should be corrected: Title page - Abstract & Keywords - Introduction - Results - Discussion - Methods - Data Availability - Acknowledgements - Disclosure and Competing Interests Statement - References - Figure Legends - Table(s)- Expanded View Figure Legends.
13. Cells are reused between Figure 4G and Appendix Fig S8 A,B,C, but this is not mentioned in the figure legend. Please explicitly note this reuse in the Appendix Figure S8 legend.
14. Source data:
 - Source data for Figure 3E and Figure 4E are missing- please clarify this in the SD checklist;
 - Source data files should be organized in a one-figure-per-folder structure and uploaded as .zip files. For example, all source data files for Figure 1 should be saved in a single folder, zipped, and uploaded as "SD_Figure_1.zip."
 - For EV and/or appendix figures, ZIP all source data together.
 - Completed SD checklist should be uploaded as Related Manuscript File.

15. Please address the following issues in figure legends :

- Please indicate what */ **/ ***/**** represents; if this represents p value(s), please the exact p value in the legend(s) of figure(s) 1C
- Please note that the exact p values are not provided in the legends of figures 4E, F; EV2 A
- Please indicate the statistical test used for data analysis in the legends of figures 2E, 3E, F; 4B, E, F, H, I, J; 5G, EV2 D
- Please note that the box plots need to be defined in terms of minima, maxima, centre, bounds of box and whiskers, and percentile in the legends of figures 2E, 3B, H, I; 5B, EV2 A, EV5A-D
- Please note that information related to n is missing in the legends of figures 1C, 2E, 3F, 5A; EV2 A, EV5 D.
- Please note that the error bars are not defined in the legends of figures 1C, 4B, E, F, H, I, J; 5D, F, G
- Please note that the measure of center for the error bars needs to be defined in the legend of figure 5A
- Please note that the scale bar needs to be defined for figure 4D.

When you resubmit your manuscript, please download our CHECKLIST (<https://bit.ly/EMBOPressAuthorChecklist>) and include the completed form in your submission. *Please note* that the Author Checklist will be published alongside the paper as part of the transparent process (<https://www.embopress.org/page/journal/17444292/authorguide#transparentprocess>)

Click on the link below to submit your revised paper.

Yours sincerely,
Jingyi

Jingyi Hou, PhD
Senior Editor
Molecular Systems Biology

If you do choose to resubmit, please click on the link below to submit the revision online before 6th Apr 2025.

*** PLEASE NOTE *** As part of the EMBO Press transparent editorial process initiative (see our Editorial at <https://dx.doi.org/10.1038/msb.2010.72> , Molecular Systems Biology will publish online a Review Process File to accompany accepted manuscripts. When preparing your letter of response, please be aware that in the event of acceptance, your cover letter/point-by-point document will be included as part of this File, which will be available to the scientific community. More information about this initiative is available in our Instructions to Authors. If you have any questions about this initiative, please contact the editorial office (msb@embo.org).

All editorial and formatting issues were resolved by the authors.

2nd Apr 2025

Manuscript number: MSB-2025-12851R

Title: Proteomics of colorectal tumors identifies the role of CAVIN1 in tumor relapse

Dear Jesper,

Thank you again for sending us your revised manuscript. We are now satisfied with the modifications made and I am pleased to inform you that your paper has been accepted for publication.

Sincerely,
Jingyi

Jingyi Hou, PhD
Senior Editor
Molecular Systems Biology
